# Bridging the High-Frequency Data Gap: A Millisecond-Resolution Network Dataset for Advancing Time Series Foundation Models

## Abstract

Time series foundation models (TSFMs) require diverse, real-world datasets to adapt across varying domains and temporal frequencies. However, current large-scale datasets predominantly focus on low-frequency time series with sampling intervals, i.e., time resolution, in the range of seconds to years, hindering their ability to capture the nuances of high-frequency time series data. To address this limitation, we introduce a novel dataset that captures millisecond-resolution wireless and traffic conditions from an operational 5G wireless deployment, expanding the scope of TSFMs to incorporate high-frequency data for pre-training. Further, the dataset introduces a new domain, wireless networks, thus complementing existing more general domains like energy and finance. The dataset also provides use cases for short-term forecasting, with prediction horizons spanning from 100 milliseconds (1 step) to 9.6 seconds (96 steps). By benchmarking traditional machine learning models and TSFMs on predictive tasks using this dataset, we demonstrate that most TSFM model configurations perform poorly on this new data distribution in both zero-shot and fine-tuned settings. Our work underscores the importance of incorporating high-frequency datasets during pre-training and forecasting to enhance architectures, fine-tuning strategies, generalization, and robustness of TSFMs in real-world applications.

## 1 Introduction

Foundation models (FMs) have significantly enhanced machine learning (ML) by utilizing large-scale pre-training on diverse datasets, enabling them to generalize across a wide array of tasks and domains (Thakur, 2024). Recently, time series foundation models (TSFMs) have attracted more interest due to their capability to handle complex temporal tasks, with a particular focus on generalizing across varying time scales and domains, including forecasting, anomaly detection, and classification (Liang et al., 2024). However, developing effective TSFMs requires access to datasets that capture diverse real-world scenarios at varying frequencies and across different domains. The blue dots in Fig. 1 demonstrate that the existing benchmark datasets predominantly focus on low-frequency time series with sampling intervals in the range of seconds to years.

Hence, the focus of this paper is to develop and benchmark a high-frequency wireless network dataset in the millisecond resolution by comparing the performance of TSFMs with shallow machine learning models to enable new architectures and fine-tuning strategies that can extend to high-frequency wireless network data use cases and potentially provide generalizable and diverse characteristics that can improve the accuracy of TSFMs on existing datasets as well.

The main contributions of this paper and dataset are: (1) Extending the scope of pre-training and generalizability for state-of-the-art TSFMs by providing a dataset at millisecond resolution (Fig. 1). (2) Introduction of a new domain, namely, wireless networks, to the existing domains of open datasets (Fig. 2). (3) Applications with short-term forecasting, with prediction horizons spanning from 100 milliseconds (1 step) to 9.6 seconds (96 steps) (Fig. 3).

The rest of the paper is organized as follows. Related work is discussed in Section 2. Section 3 provides a detailed description of the 5G network data, and its characteristics. Section 4 presents the

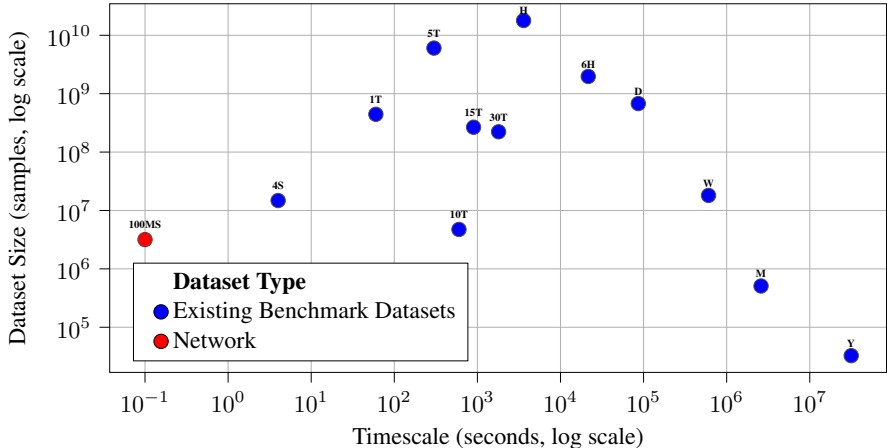

Figure 1: Comparison of timescales and dataset sizes for standard existing datasets used for pre-training (Table 14 in (Aksu et al., 2024)) as compared with the new benchmark. The red dot represents the new dataset that is introduced in this paper.

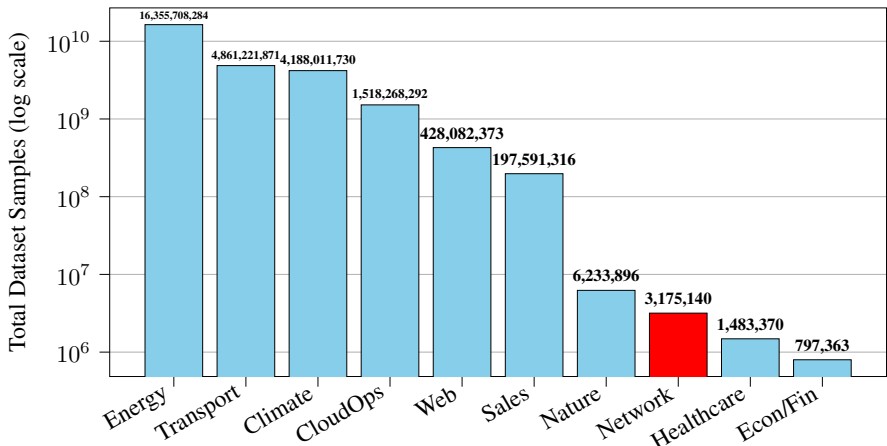

Figure 2: Comparison of existing domains for pre-training (Table 14 in (Aksu et al., 2024)) with the new benchmark. The red bar represents the new dataset that is introduced in this paper.

details of models benchmarked, including experimental evaluation and analysis. Section 5 outlines the ablation study. Finally, in Section 6, we conclude and provide directions for future research.

## 2 RELATED WORK

Time Series Foundation Models (TSFMs) have surged in recent years, with their architectures continually evolving to achieve improved performance in both zero-shot and fine-tuned scenarios. Notably, several TSFMs have garnered widespread attention within the community, including Chronos (Ansari et al., 2024), TTM (Ekambaram et al., 2024), Moirai (Woo et al., 2024), TimesFM (Das et al., 2024), and Time-MOE (Xiaoming et al., 2025). These models can be broadly categorized into two distinct classes: transformer-based and non-transformer-based architectures (Liang et al., 2024). Our work complements these developments by introducing a high-frequency, real-world dataset from a novel domain (wireless networks), which provides an additional and challenging benchmark for evaluating the robustness and adaptability of TSFMs.

Transformer-based TSFMs largely follow established self-supervised (e.g., Moirai) or supervised transformer frameworks (e.g., TimeXer), which have garnered significant recognition within the field. In contrast, non-transformer-based TSFMs leverage alternative machine learning models such

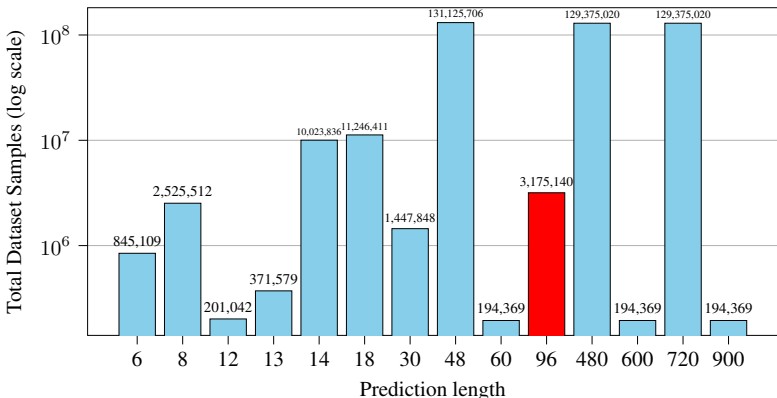

Figure 3: Comparison of prediction lengths of standard test data (Table 2 in (Aksu et al., 2024)) as compared with the new benchmark. The red bar represents the new dataset that is introduced in this paper.

as Multi-Layer Perceptron (MLP) and Convolutional Neural Networks (CNN) (e.g., TTMs). More recent efforts have also focused on enhancing diffusion-based methods (Kollovieh et al., 2023; Su et al., 2025) for modeling and generating data of different characteristics, which is crucial for generative time series forecasting. Furthermore, to address statistical heterogeneity in time series foundation model training and ensure robust generalization, a decentralized cross-domain model fusion approach, as Federated Learning (FL), has been explored in (Chen et al., 2025).

The successful deployment of these TSFMs for accurate zero-shot forecasting relies on the development of pre-trained models that have undergone extensive training on datasets characterized by diverse patterns and resolution properties. This emphasis on data diversity is critical, as it enables TSFMs to exhibit generalizability across a wide range of scenarios and capture complex temporal dynamics with enhanced accuracy. Notably, prior research has underscored the importance of resolution and domain diversity in pre-trained models for optimizing performance (e.g., Section 4 in (Ansari et al., 2024) for Chronos and Section 4.9 and Fig. 3 in (Ekambaram et al., 2024) for TTM.

In practice, a range of open datasets is available for TSFMs, which collectively provide the necessary heterogeneity to ensure that these models generalize effectively to out-of-domain datasets and real-world applications. Specifically, popular datasets such as those from Monash (Godahewa et al., 2021), LIBCITY (Wang et al., 2021), and the UCI Machine Learning archive (Asuncion et al., 2007) have become foundational in pre-training TSFMs and are widely utilized for assessing model performance. These datasets not only serve as data for pre-trained models but also enable out-of-domain testing of pre-trained models when a subset of the datasets are not considered for pre-training. We position our dataset as a complementary resource to these existing open datasets, specifically targeting the gap for millisecond-level time series from communication networks for both training and out-of-domain evaluation of TSFMs. Our dataset directly addresses this need for diversity by introducing a previously underrepresented domain with very fine temporal granularity, thereby contributing to a better understanding of the generalization capabilities of TSFMs when applied to high-frequency wireless data.

This paper provides a benchmark dataset that can fill the critical gap for high-frequency data for TSFMs. In contrast to other high-frequency datasets, our network dataset provides carefully curated use cases for univariate and multivariate forecasting problems ideally suited for TSFMs, along with an initial benchmark study on this dataset.

## 3 DATASET

### 3.1 DATASET OVERVIEW

We utilize a time series dataset of 5G Radio Access Network (RAN) Performance Measurements (PMs) collected from a real-world deployment of a 5G Open Radio Access Network (O-RAN)

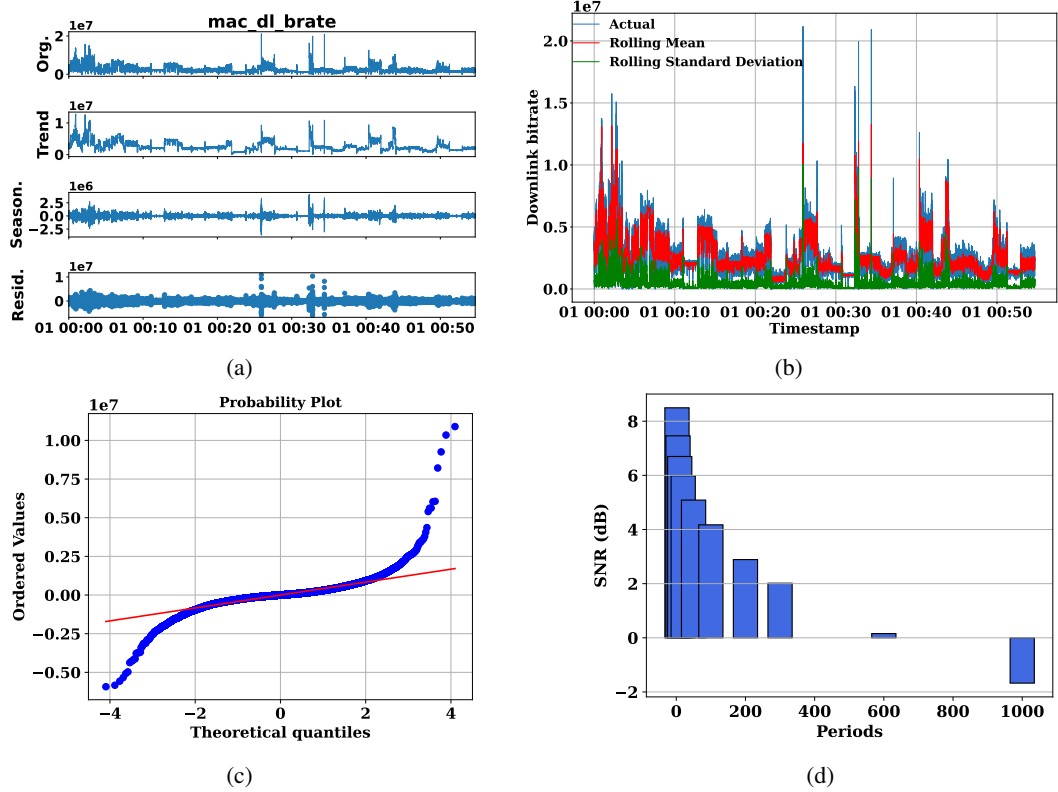

Figure 4: Target variable (Downlink Bitrate; mac_dl_brate): (a) STL decomposition, (b) Rolling mean and standard deviation, (c) Residual Q-Q, (d) Signal-to-Noise Ratio (dB).

Table 1: Summary of STL Decomposition of all datasets.

| Dataset | STL Decomposition | | |
|---|---|---|---|
| | Trend | Seasonality | Residuals |
| Network | Unstable, step-like shifts. | Weak short-term periodic patterns. Hidden by noise. | Sharp spikes. Bursts of noise. Lots of unpredictable variation. |
| ETTh1 | Mostly steady with small rises and falls. | Small, regular repeating pattern. | Tiny random changes. |
| Electricity | Remains steady throughout. | Strong repeating pattern. | Occasional bursts of noise. |
| Weather | Almost flat but interrupted by sudden sharp spikes. | No seasonality. | Mostly small, but with rare sudden jumps. |
| Traffic | Slowly increasing trend over time. | Strong, regular repeating pattern. | Small random changes. |

within the OpenIreland testbed. O-RAN introduces a modular and open architecture that decomposes the traditional monolithic RAN into standardized, interoperable components (i.e, the Central Unit (CU), Distributed Unit (DU), and Radio Unit (RU)) facilitating multi-vendor deployments and software-driven control. Central to O-RAN's programmability is the near-Real-Time RAN Intelligent Controller (near-RT RIC), which enables rapid, feedback-driven network optimization.

The data was captured using software-defined radios (Ettus USRPs) configured as a base station and multiple user equipments (UEs). To simulate diverse real-world usage, the setup incorporated various mobility profiles (static, pedestrian, car, bus, and train) and generated traffic from both benign applications (web browsing, VoIP, IoT, and video streaming) and malicious activities (DDoS-Ripper,

DoS-Hulk, PortScan, Slowloris). PMs were collected at the base station side and span a broad set of physical and medium access control layer features, including the Channel Quality Indicator (CQI), Modulation and Coding Scheme (MCS), Noise ratio interference (SINR), Signal strength (RSSI), buffer occupancy, and packet delivery statistics. In the dataset, each UE is associated with a unique identifier, denoted as *ue_ident*, which serves to distinguish individual UEs across all collected traces. This identifier remains consistent for a given UE, regardless of the mobility pattern or traffic class associated with its traces. The resulting dataset enables temporal modeling of RAN dynamics under realistic operational conditions.

This data context is particularly well-suited for very short-term forecasting, where the goal is to predict network states (e.g., throughput, channel quality, traffic class) over a short horizon ranging from milliseconds to a few seconds. Such forecasting enables predictive control strategies in scenarios characterized by rapid fluctuations in load, mobility, or interference (see Section 3.2 for dataset characteristics). Short-term throughput predictions enhance scheduling efficiency and application-level rate control, especially in latency-sensitive services like cloud gaming or interactive video. Forecasting CQI, for example, allows the network to proactively steer users to cells with better anticipated radio conditions, support load-aware handovers, and preemptively adjust adaptive bitrate algorithms for video streaming. Likewise, anticipating traffic class transitions supports early enforcement of QoS policies, dynamic resource allocation (e.g., in network slicing), and intrusion detection mechanisms capable of identifying malicious activity before it significantly degrades the service.

## 3.2 DATASET CHARACTERISTICS

While Section 3.1 provides a broad overview of the 5G network dataset, our analysis and experiments are carried out on a carefully filtered subset of the data. We filter the raw data on the basis of the mobility pattern and benign traffic class. In particular, the *static* mobility pattern for the *video streaming* traffic class. Therefore, the results presented here represent the characteristics of the filtered dataset rather than those of the complete dataset.

The time series of the 5G network demonstrates several important characteristics. Fig. 4a shows the STL (Seasonal and Trend decomposition using Loess) of the time series, which separates the original data (labeled Org. in Fig. 4a) into distinct structural components, i.e., the trend, seasonal and residual components. Here, the trend component reflects the underlying structure of the series; however, it appears unstable, as characterized by step-like shifts rather than a smooth trajectory. The seasonal component captures only weak short-term periodic patterns, which are easily obfuscated by the stronger irregular behavior in the data. The residual component contains the remaining variability, including sharp spikes and bursts of endogenous noise that cannot be explained by trend or seasonality. Similarly, as illustrated in Fig. 4b, both the rolling mean and the standard deviation are observed to change substantially over time, confirming that the process is non-stationary and heteroskedastic. This means that the statistical properties of the data are not constant. The data exhibit extreme outlier events that are more prominent in specific time periods than in random events throughout the series. The autocorrelation analysis (see Section 9) reveals a strong temporal persistence with slow decay, confirming the clustering of extreme events observed in the data. In Fig. 4c, the residuals deviate strongly from the reference line, particularly in the tails, indicating a heavy-tailed distribution. Finally, the signal-to-noise ratio (SNR) analysis in Fig. 4d provides a quantitative view of this instability. The SNR values highlight that the series is dominated by short-term periodic structures (high SNR in periods 2-20), while medium-term cycles exist but are weaker, and long-term seasonality is essentially absent (SNR nears to zero and even negative beyond period 600). Overall, the time series is mostly influenced by short-term changes, bursts of volatility and clustered anomalies, rather than stable long-term trends.

Next, we provide a summary of the overview on the comparison between our 5G network dataset and other common pre-trained datasets (further experimental details are presented in Appendix A.4). The pre-trained datasets used for comparison are: **ETTh1 (Zhou et al., 2021)** is an hourly subset of the Electricity Transformer Temperature (ETT) dataset, containing two years of transformer oil temperature and related power load data from two counties in China. **Electricity (Wu et al., 2021)** dataset contains the hourly electricity consumption(in kWh) from 321 clients, recorded between 2012 and 2014. **Weather (Wu et al., 2021)** data from 2020 in Germany, recorded every 10 minutes, with 21 indicators such as air temperature, humidity, and wind speed. **Traffic (Wu et al., 2021)** is a collection of hourly road occupancy rates (0–1) from sensors on San Francisco Bay Area freeways,

Table 2: Features used in multivariate setting.

| Feature | Description |
|---------|-------------|
| CQI | Channel Quality Indicator |
| MCS | Modulation and Coding Scheme |
| pkt ok | Number of packets sent |
| pkt nok | Number of packets dropped |

collected by the California Department of Transportation between 2015 and 2016. Table 1 summarizes the key differences among the datasets based on their STL decomposition, highlighting that our dataset is notably different due to its unstable trend, weak seasonality, and spiky residuals. Appendix A.4 includes other data characteristics, such as temporal dependencies, and statistical variability.

## 4 BENCHMARK

In this section, we provide a comprehensive analysis of the benchmarked models (as explained in 4.1) for the considered target variable *downlink bitrate* (**bitrate**) in the 5G network dataset. In the multivariate setting, all considered models use four input features, with descriptions provided in Table 2. Section 4.3 provides implementation details, including the data processing pipeline, that reflects our consideration of only a subset of data to illustrate the impact of this high-frequency dataset.

### 4.1 MODELS BENCHMARKED

We selected three state-of-the-art tree-based ensemble models: Random Forest (RF) (Breiman, 2001), implemented using Scikit-learn, eXtreme Gradient Boosting (XGBoost, hereafter XGB) (Chen & Guestrin, 2016), and Adaptive Random Forest (ARF) (Gomes et al., 2018) implemented using the River library. As an additional online baseline, we included a simple incremental linear regression model, Online LR (OLR) (Ouhamma et al., 2021), also implemented using the River library. Similarly, we selected a non-parametric baseline, referred to as naive forecast (Naive) (Beck et al., 2025), for a fair evaluation on high-frequency data.

In addition, we evaluated three time series foundation models (TSFMs): TinyTimeMixer (TTM) (Ekambaram et al., 2024), Chronos (Ansari et al., 2024), and Lag-Llama (Rasul et al., 2023), each specifically designed for time series forecasting. TTM is an extremely light-weight pre-trained model, with effective transfer learning capabilities based on the light-weight TSMixer architecture. Likewise, Chronos is a language modeling framework for time series for pre-trained probabilistic time series models. In this work, we specifically adopted the Chronos-bolt-small variant (46M parameters) as the representative Chronos model for our experiments. Lag-Llama is a general-purpose foundation model for univariate probabilistic time series forecasting based on a decoder-only transformer architecture that uses lags as covariates.

### 4.2 SYSTEM SPECIFICATION

The experiments are carried out on a local machine with the following hardware and software specifications: **Operating System:** Microsoft Windows 10 Enterprise, Version 22H2; **Processor:** 11th Gen Intel® Core™ i7-1165G7 CPU @ 2.80 GHz with 4 cores and 8 threads; **Memory:** 32 GB RAM.

### 4.3 IMPLEMENTATION DETAILS

*Pre-processing:* During data pre-processing, we changed the time resolution of our dataset by converting the original millisecond-level observations into 100-millisecond intervals. This choice reflects practical constraints in O-RAN networks, where collecting performance measurements at every millisecond would impose excessive overhead. For shallow models, input sequences are constructed using a sliding-window approach, where past observations within a fixed window are used to predict future target values. For TSFMs, we follow the original implementation protocols described

Table 3: Parameters used in model training.

| Parameter | Univariate | Multivariate |
|---|---|---|
| n_models | 10 | 20 |
| max_features | None | 0.5 |
| grace_period | 50 | 100 |
| max_depth | None | 5 |

(a) Hyper-parameters specific to ARF.

| Parameter | Value |
|---|---|
| Target variable | Downlink bitrate |
| No. of features | 4 |
| Mobility pattern | Static |
| Past observations | 5 |
| Prediction horizon | 96 |
| Train set:Test set | 80:20 |

(b) Common parameters for all shallow models.

Table 4: Performance metrics of benchmarked models.

| Model | Univariate | | Multivariate | |
|---|---|---|---|---|
| | RMSE | MAE | RMSE | MAE |
| RF | $0.0344 \pm 0.0001$ | $0.0227 \pm 0.0001$ | $0.0342 \pm 0.0001$ | $0.0226 \pm 0.0001$ |
| XGB | $0.0354 \pm 0.0001$ | $0.0232 \pm 0.0001$ | $0.0354 \pm 0.0001$ | $0.0231 \pm 0.0001$ |
| ARF | $\mathbf{0.0270 \pm 0.0002}$ | $0.0189 \pm 0.0001$ | $\mathbf{0.0175 \pm 0.0007}$ | $\mathbf{0.0130 \pm 0.0005}$ |
| Naive | $0.0418 \pm 0.0000$ | $0.0240 \pm 0.0000$ | $0.0418 \pm 0.0000$ | $0.0240 \pm 0.0000$ |
| OLR | $0.0551 \pm 0.0000$ | $0.0308 \pm 0.0000$ | $0.0555 \pm 0.0000$ | $0.0310 \pm 0.0000$ |
| TTM (Zero-shot) | $0.0359 \pm 0.0000$ | $0.0230 \pm 0.0000$ | $0.0359 \pm 0.0000$ | $0.0230 \pm 0.0000$ |
| TTM (Fine-tuning) | $0.0371 \pm 0.0015$ | $0.0237 \pm 0.0011$ | $0.0393 \pm 0.0007$ | $0.0250 \pm 0.0004$ |
| Chronos (Zero-shot) | $0.0313 \pm 0.0000$ | $0.0185 \pm 0.0000$ | $0.0273 \pm 0.0000$ | $0.0181 \pm 0.0000$ |
| Chronos (Fine-tuning) | $0.0281 \pm 0.0000$ | $\mathbf{0.0178 \pm 0.0000}$ | $0.0253 \pm 0.0000$ | $0.0176 \pm 0.0000$ |
| Lag-Llama (Zero-shot) | $0.0617 \pm 0.0002$ | $0.0384 \pm 0.0001$ | - | - |
| Lag-Llama (Fine-tuning) | $0.0474 \pm 0.0039$ | $0.0268 \pm 0.0009$ | - | - |

in their respective papers. The prediction horizons range from 1 millisecond up to 9.6 seconds. Short-term horizons are often straightforward, as the target variable (i.e., bitrate) tends to remain stationary across very small timescales. In contrast, longer horizons provide more meaningful insights, enabling applications such as video streaming to anticipate changes in bitrate and proactively adjust parameters like encoding level. These long horizon forecasts are valuable both for adapting Quality of Service (QoS) and for estimating the stability of the bitrate, that is, how frequently it is expected to change.

*Model parameters:* Table 3 summarizes the parameters used during model training. For common parameters shared across all models, offline experiments were conducted to select optimal values based on prediction accuracy, ensuring fair benchmarking conditions. Both RF and XGB models used these optimized common parameters along with their respective default model-specific hyper-parameters without additional tuning. For the ARF model, the same optimized common parameters were used, and model-specific hyper-parameter tuning was performed for multivariate setting using random search methodology, with parameter ranges detailed in Table 3a. The best performing Root Mean Square Error (RMSE)-based ARF configuration was selected for the final evaluation. Furthermore, the prediction horizon for all models is set at 96 steps, with each step representing a 100-millisecond interval; which corresponds to predicting the next 9.6 seconds (9600 ms). The dataset is divided into 80% training and 20% testing, preserving temporal order. As the data for each user are sequential and not mixed, this split naturally keeps the sequence of each user intact, preventing data leakage from future observations into training.

*Model training:* For RF and XGB, we utilized Scikit-learn's *MultiOutputRegressor* wrapper to enable direct multi-step forecasting. For TSFMs, we follow the original implementation protocols described in their respective papers. Each model was trained with three different random seeds (42, 99, and 123) to ensure reproducibility and to assess variability in performance.

*Post-processing:* Both Chronos and Lag-Llama are trained to predict a fixed length horizon **H** from a given context window. By default, these models produce forecasts only for the final prediction window of each series and skip series that do not meet the minimum context length. This default

evaluation framework differs from shallow models that generate forecasts for every test sample. To ensure a consistent comparison across models, we implemented a rolling evaluation procedure for both Chronos and Lag-Llama. Specifically, starting with each timestamp **t**, we provide the model with all historical data available up to **t** and generate the next steps **H**. We then slide the starting point forward by one time step and repeat the prediction until the end of the series. This produces overlapping multi-step forecasts aligned with each test timestamp, allowing direct comparison with the shallow models.

To extend Chronos, which is inherently a univariate model, to the multivariate setting, we use AutoGluon-TimeSeries (AG-TS) covariate regressors (Shchur et al., 2023). The covariate regressor is a tabular model trained on known covariates and static features to predict the target at each time step. Its predictions are subtracted from the target series, and Chronos then forecasts the residuals. For each rolling window, we create a future covariate table that matches the next **H** time steps immediately following the end of the current window. This table contains the values of the exogenous variables for those steps, allowing Chronos to use both the historical target and the future covariates to generate accurate forecasts.

## 4.4 RESULTS

In this section, we evaluate the performance of shallow models and TSFMs in both univariate and multivariate settings. Table 4 presents the performance of the benchmarked shallow models and TSFMs, evaluated using RMSE and Mean Absolute Error (MAE). In both settings, ARF consis-

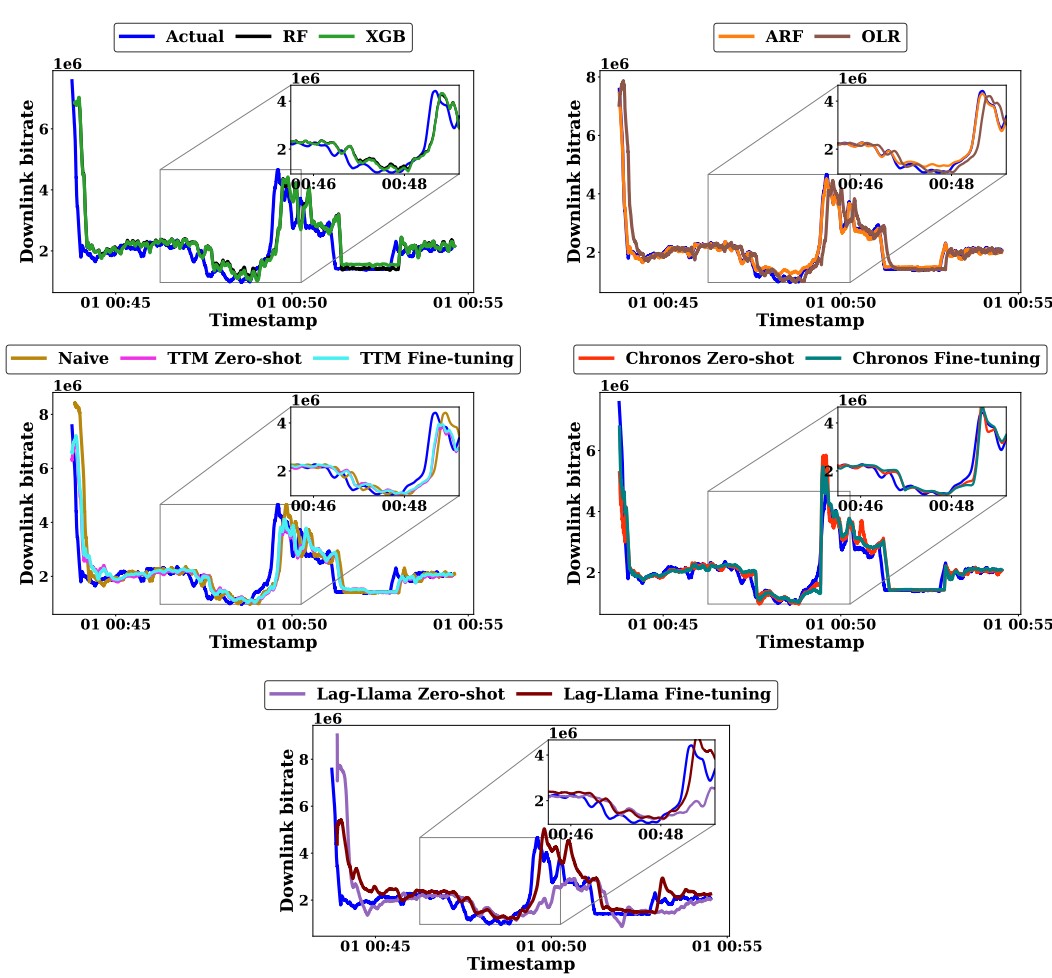

Figure 5: Actual v.s. Predicted bitrate values in a Univariate setting.

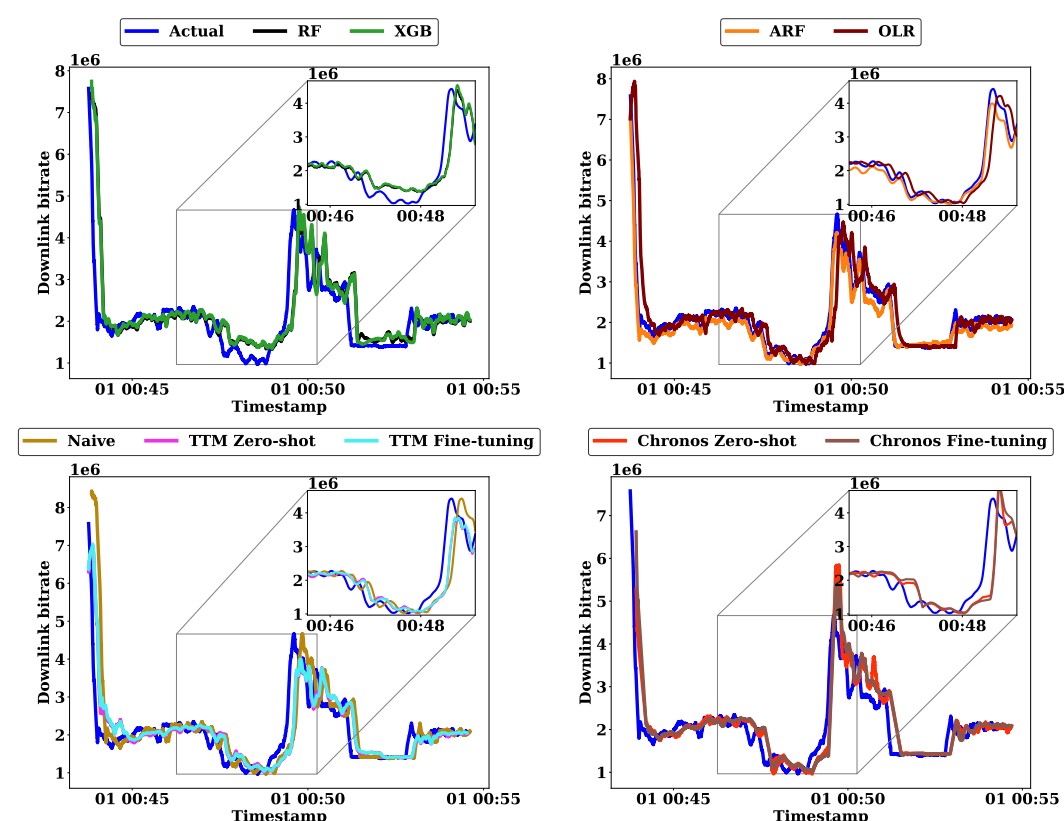

Figure 6: Actual v.s. Predicted bitrate values in a multivariate setting.

Table 5: Performance Metrics for Different Fine-Tuning strategies for TTM.

| Fine-tuning Strategy | RMSE | MAE |
|---|---|---|
| Head-only fine-tuning | 0.0413 | 0.0270 |
| Adapter-based fine-tuning | 0.0522 | 0.0334 |

tently outperforms the other shallow models and TSFMs. The performance gain is consistent with the data characteristics observed in Section 3.2; our 5G network dataset is dominated by irregular spikes, step-like changes, and lack of stable seasonality. Static models such as RF or XGB struggle in performance because they assume that the training distribution does not change over time, leading to poor generalization when sudden data shifts occur. While the Online LR baseline, despite updating incrementally, cannot fully capture the complex, non-linear dynamics in the data. Similarly, TSFMs performance degrades due to a shift in data distribution in the zero-shot scenario, as these pre-trained models are trained only on low-frequency data, limiting their ability to capture high-frequency dynamics with unpredictable spikes and irregular patterns. Based on the results it can be seen that even after fine-tuning and further hyper-parameter tuning (see Section A.5.3) on our dataset, the performance of TSFMs remains suboptimal, as they fail to generalize effectively. In contrast, ARF is designed to handle concept drift by dynamically updating its ensemble of trees as new patterns appear. This allows it to quickly adapt to data distribution changes and maintain predictive accuracy even in the presence of strong irregularities. While it is observed that Chronos offers a competitive performance in the univariate setting, ARF outperforms Chronos in the multivariate setting.

The performance of these models is more clearly reflected in Fig. 5 and Fig. 6. We observe that ARF follow the curve/trend of the **bitrate** much better than the other shallow models and TSFMs. For the purpose of visualization, we average the actual and predicted values for each test sample.

Table 6: Performance metrics of benchmarked models with the increasing temporal resolution.

| Temporal Resolution | Prediction Horizon | ARF | | TTM Zero-shot | | TTM Fine-tuning | | Chronos Zero-shot | |
|---|---|---|---|---|---|---|---|---|---|
| | | RMSE | MAE | RMSE | MAE | RMSE | MAE | RMSE | MAE |
| 100 ms | 96 | **0.0457** | **0.0262** | 0.0765 | 0.0434 | 0.0743 | 0.0421 | 0.0622 | 0.0338 |
| 200 ms | 48 | **0.0471** | **0.0267** | 0.0870 | 0.0499 | 0.0880 | 0.0496 | 0.0740 | 0.0389 |
| 500 ms | 20 | **0.0398** | **0.0218** | 0.0855 | 0.0490 | 0.0894 | 0.0542 | 0.0711 | 0.0372 |
| 1000 ms | 10 | **0.0297** | **0.0176** | 0.0856 | 0.0500 | 0.0856 | 0.0500 | 0.0580 | 0.0326 |
| 2000 ms | 5 | **0.0289** | **0.0169** | 0.0880 | 0.0527 | 0.0915 | 0.0584 | 0.0671 | 0.0354 |
| 3000 ms | 4 | **0.0289** | **0.0185** | 0.1049 | 0.0618 | 0.1061 | 0.0638 | 0.0860 | 0.0443 |

## 5 ABLATION STUDY

### 5.1 FINE-TUNING STRATEGIES FOR TTM

In this section, we analyze how different fine-tuning strategies affects the performance of TTM. We explore two different fine-tuning strategies: **(i) Head-only fine-tuning** (Ekambaram et al., 2024), where we freeze the entire backbone and decoder and only train the final prediction head, **(ii) Adapter-based fine-tuning** (Houlsby et al., 2019), where we incorporate lightweight MLP adapter modules inside the mixer blocks while keeping the original TTM weights frozen. Recent works on fine-tuning TSFMs (Tomar et al.) has shown that even widely used Parameter-Efficient Fine-Tuning (PEFT) methods like Low-Rank Adaptation (LoRA) do not consistently improve the performance of TSFMs. Our findings in Table 5 aligns with this observation; even though both the fine-tuning strategies are architecturally compatible with TTM, their performance is worse as compared to default TTM fine-tuning approach.

### 5.2 TEMPORAL RESOLUTION

In this section, we evaluate the performance of ARF and TSFMs in a multivariate setting. We analyze the effect of increasing the temporal resolution of the data on the performance of both ARF and TSFMs, performing fine-tuning only for TTM because of its computational efficiency. The prediction horizon is fixed at 9.6 seconds for all temporal resolutions. Specifically, we evaluate these models on a newly filtered data; *pedestrian* mobility pattern for the **video streaming** traffic class, to highlight that the characteristics of our dataset differ from those of the pre-trained datasets. Table 6 shows the performance of ARF and TSFMs. Notably, increasing the temporal resolution does not improve the performance of TSFMs. In contrast, ARF consistently outperforms TSFMs at each resolution, as higher temporal resolution reduces noise and improves its predictions. This indicates that TSFMs perform poorly not only because of temporal resolution (i.e., high frequency), but also due to the inherent characteristics of our data.

## 6 CONCLUSION AND FUTURE WORK

We present a novel high-frequency time series dataset capturing millisecond-resolution measurements from real-world wireless network. This dataset fills a critical gap in existing large-scale resources, which largely lack fine-grained, real-time wireless network data. Our experiments reveal the limitations of current TSFMs and highlight the need to incorporate diverse, high-resolution datasets during pre-training to improve generalization. In the future, we will use this dataset for the use case of anomaly detection and transfer learning across various mobility profiles.

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

# A APPENDIX

## A.1 USE CASES

The dataset enables a wide range of learning tasks that support more adaptive Radio Access Network (RAN) behavior, particularly within O-RAN systems where short-term predictions and rapid classifications can guide near-real-time control. Its millisecond-resolution measurements, combined with detailed PHY- and MAC-layer indicators and labels for traffic type and mobility class, allow the design of regression models that forecast throughput, channel quality, and link reliability over horizons from a few milliseconds to several seconds. Such predictions can inform scheduling decisions at the DU, guide proactive MCS and power adjustments, improve rate control for latency-sensitive applications, and support mobility steering by anticipating future channel degradation for fast-moving users. The temporal characteristics of the dataset, including irregular bursts and heavy-tailed dynamics, make it well suited for evaluating predictive approaches in environments where rapid fluctuations dominate.

The dataset also supports classification tasks involving mobility identification and traffic-type recognition. Since user movement patterns such as static, pedestrian, car, bus, and train produce distinct combinations of SINR, CQI, and bitrate variability, models trained on these traces can infer mobility behavior directly from RAN KPIs. Such inferences allow the RAN Intelligent Controller (RIC) to select mobility-aware handover strategies, tune power control settings, or commit resources more

efficiently. Traffic classification, which extends across benign and malicious flows, provides an additional line of evidence for service-awareness and security monitoring. The dataset includes benign web, VoIP, IoT, and video traffic, as well as multiple attack types such as DDoS-Ripper, DoS-Hulk, PortScan, and Slowloris. This makes it possible to detect abnormal traffic solely from network-side performance indicators, enabling security functions that do not rely on deep packet inspection.

Beyond supervised learning, the dataset's sharp spikes, volatility clusters, and inconsistent seasonal structure create strong opportunities for anomaly detection. Deviations in CQI, SINR, buffer occupancy, packet loss, or bitrate can reveal early signs of congestion, or malicious activity. Because the dataset includes both dynamic mobility patterns and diverse traffic sources, anomaly detectors built on it can be tested against conditions where network behavior changes rapidly and non-linearly. This setting mirrors real operational networks more closely than traditional low-frequency datasets and supports the design of proactive mitigation strategies within the RIC.

Finally, the dataset's combination of high-frequency time series, labelled mobility classes, and labelled traffic classes allows for multi-task learning and transfer learning studies. Models can be trained on one mobility class and evaluated on another, or jointly predict throughput while classifying user behavior. This supports research on generalization across heterogeneous RAN conditions and offers a realistic foundation for developing predictive, adaptive, and security-oriented control functions that operate within the O-RAN architecture.

## A.2 LIMITATIONS

Our current study provides valuable insights into the performance of shallow models and TSFMs for millisecond resolution wireless network data, and shows the need to utilize this dataset to enhance the generalizability and applicability of TSFM pre-training and fine-tuning capabilities. However, there are certain limitations in the study that highlight areas for potential improvement in future research. These include:

- The empirical benchmark results for shallow models such as XGBoost and Random Forest only had limited hyper-parameter tuning, whereas standard Hyperparameter Optimization (HPO) techniques could have been applied to further optimize their performance. Given the paper's primary focus on comparing benchmark performance between shallow models and TSFMs, any potential marginal improvements through HPO were deemed secondary to the main objective. Nonetheless, additional hyper-parameter tuning was performed for both TTM and Lag-Llama; however, ARF continued to outperform both models.

- Further, default implementations of the TSFMs were considered for the performance on zero-shot models. Feature engineering and data pre-processing strategies can potentially improve the performance of TSFMs but this was not considered. Since shallow models work directly on the raw data and perform reliable forecasting, the same was done for TSFMs to make the comparison fair.

- Default fine-tuning implementations were explored for each TSFM, whereas TTM was further evaluated using distinct fine-tuning strategies. However, novel techniques such as autotuning and Low-Rank Adaptation (LoRA) (Hu et al., 2022) strategies were not considered since the focus was on zero-shot and few-shot learning. Future work on ablation studies is proposed to investigate whether optimizing few-shot learning parameters can significantly enhance the performance of TSFMs.

## A.3 PERFORMANCE EVALUATION METRICS

The Root Mean Squared Error (RMSE), and Mean Absolute Error (MAE) are calculated as follows:

$$RMSE(Y_t, \hat{Y}_t) = \sqrt{\frac{1}{T} \sum_{t=1}^{T} (Y_t - \hat{Y}_t)^2}, \tag{1}$$

$$MAE(Y_t, \hat{Y}_t) = \frac{1}{T} \sum_{t=1}^{T} \left| (Y_t - \hat{Y}_t) \right|, \tag{2}$$

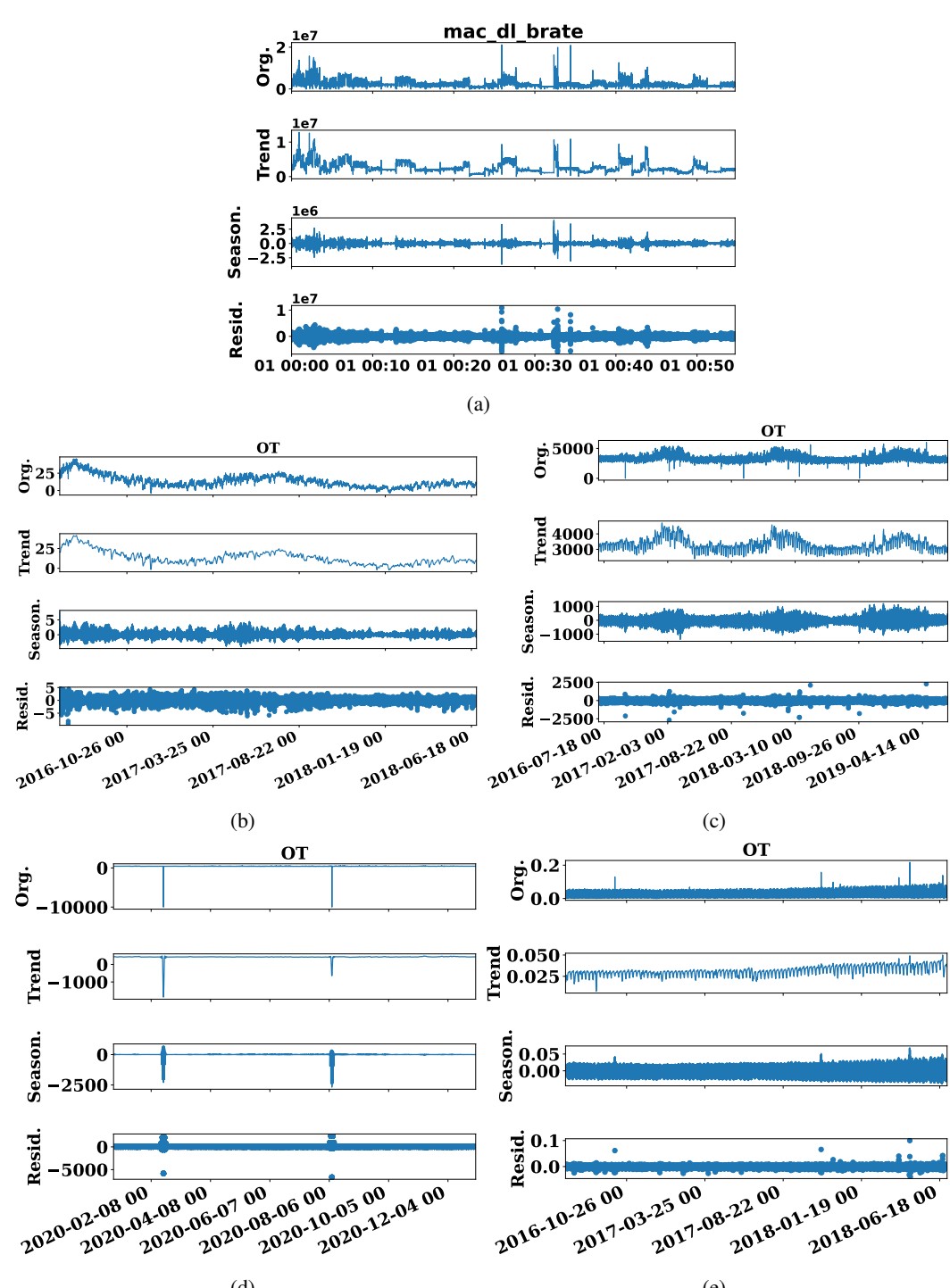

Figure 7: STL decomposition of time series: (a) Network, (b) ETTh1, (c) Electricity, (d) Weather, (e) Traffic.

where $Y_t$ and $\hat{Y}_t$ are the actual and predicted bitrate values, and $T$ is the total number of samples in the test data.

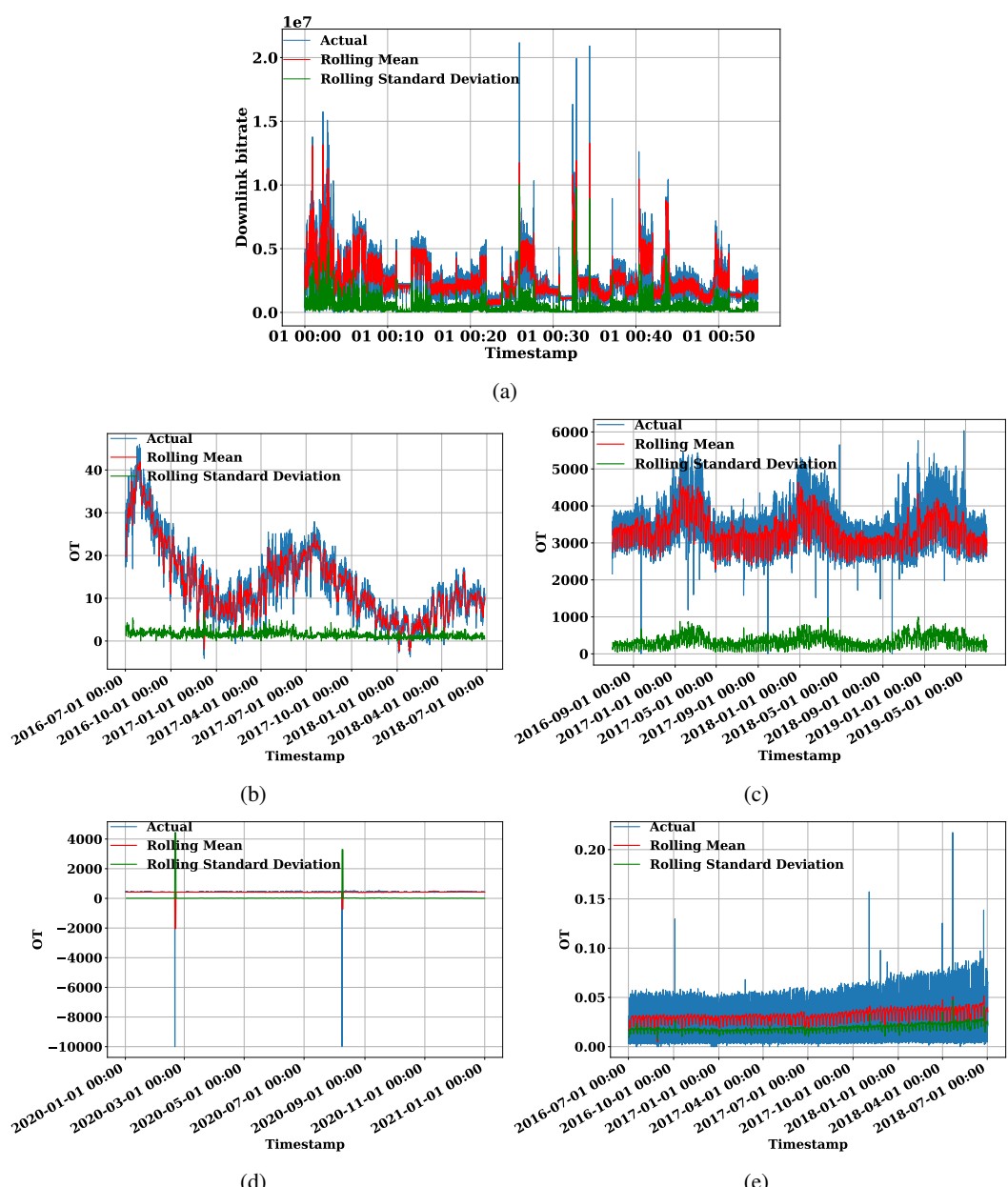

Figure 8: Rolling mean and standard deviation of time series: (a) Network, (b) ETTh1, (c) Electricity, (d) Weather, (e) Traffic.

## A.4 DATA CHARACTERISTICS COMPARISON

In this section, we compare our 5G network dataset with those used in the pre-training of TSFMs. The comparison focuses on key data characteristics, including statistical distributions, temporal dependencies, and statistical variability, as illustrated in Figs. 7, 8, 9, and 10. We compare the datasets using STL decomposition, rolling mean and standard deviation, autocorrelation (ACF), and residual QQ plots. Our 5G network data is clearly the most different; its trend shifts abruptly in steps, seasonality is weak and mostly hidden by noise, rolling statistics change suddenly, the ACF shows strong temporal persistence with slow decay, and the residual QQ plot departs strongly from normality due to sharp spikes. In contrast, the ETTh1 dataset has a mostly steady trend with mild rises and falls, small but regular seasonal cycles, stable rolling statistics, weak cyclical autocorrelation, and residuals close to normal. The Electricity dataset also remains steady in its trend but shows stronger

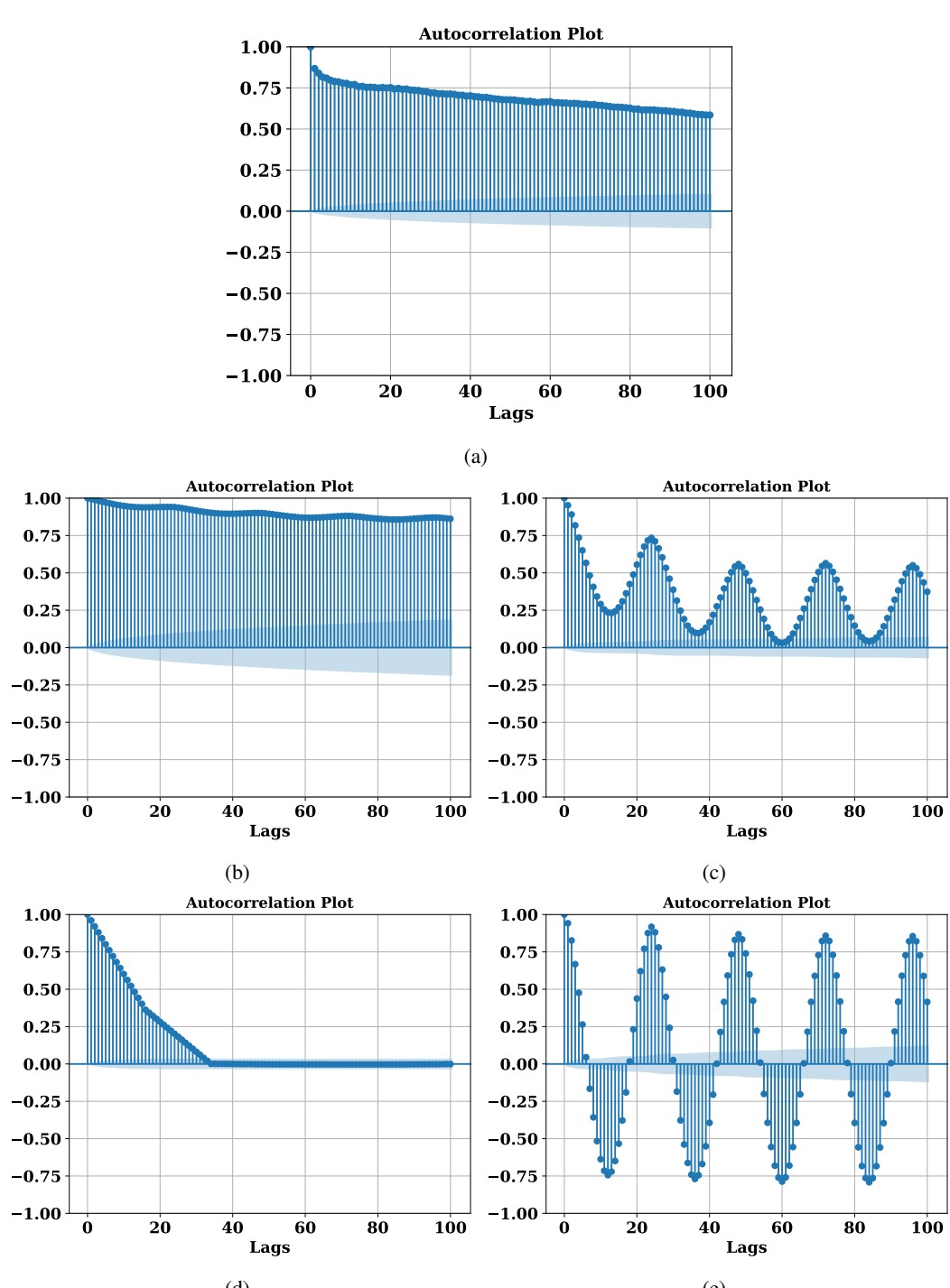

Figure 9: Autocorrelation of time series: (a) Network, (b) ETTh1, (c) Electricity, (d) Weather, (e) Traffic.

repeating seasonal patterns, its rolling mean is flat and variance is stable, clear cycles in the ACF, and residuals with occasional deviations. The Weather dataset is mostly flat with rare sharp jumps, no meaningful seasonality, sudden variance spikes in rolling statistics, weak ACF signals, and QQ plots highlighting outliers. Finally, Traffic dataset combines a smooth upward trend with strong, consistent seasonality, gradually increasing rolling mean with stable variance, clear seasonal auto-

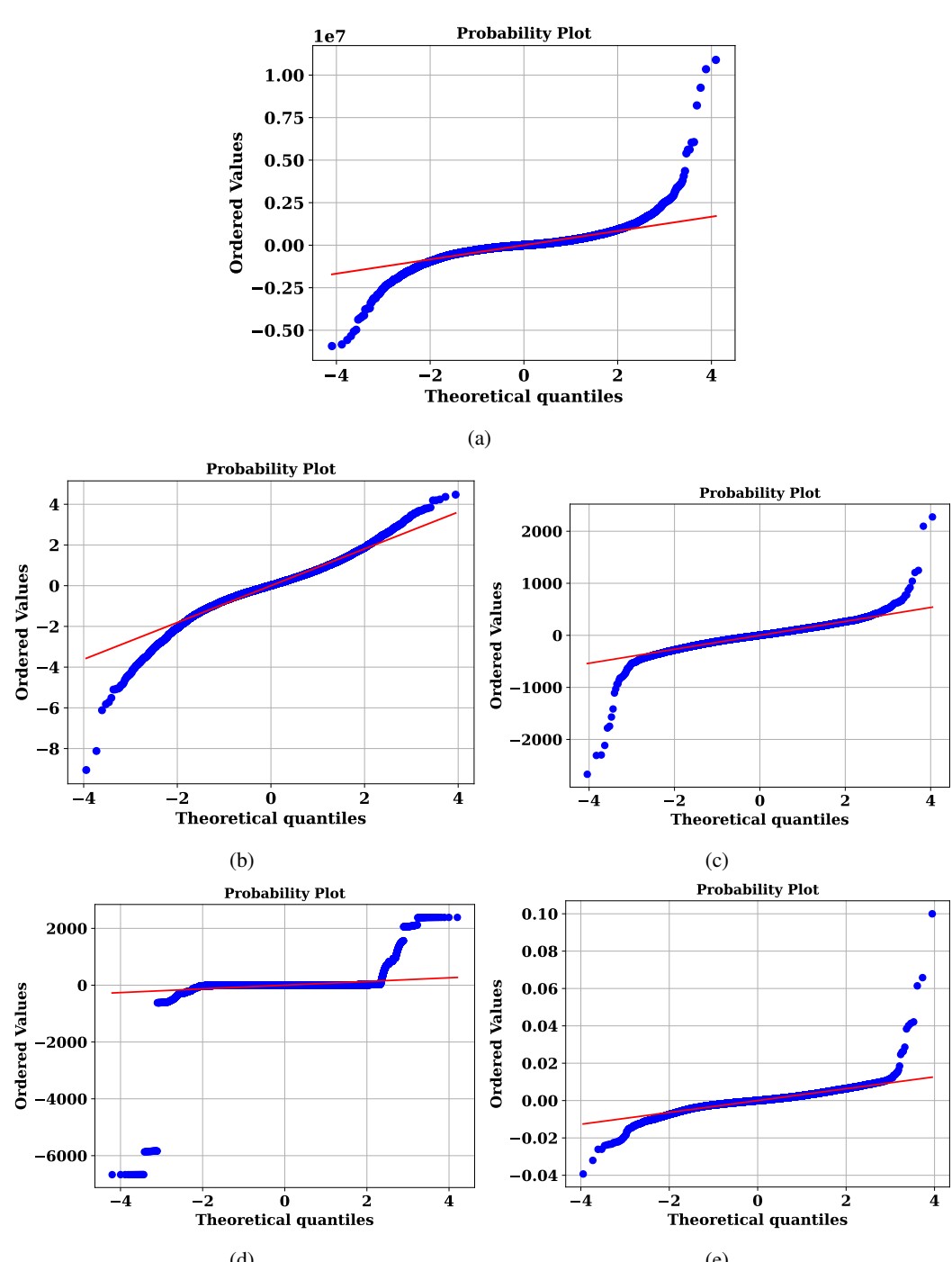

Figure 10: Autocorrelation of time series: (a) Network, (b) ETTh1, (c) Electricity, (d) Weather, (e) Traffic.

correlation, and residuals that follow normality fairly well. To conclude, our dataset differs from the others because its persistence comes from clustered extremes and abrupt shifts rather than smooth or cyclical structure, making it the least regular and most unpredictable series.

## A.5 ABLATION STUDY

### A.5.1 NUMBER OF FEATURES

Table 7 summarizes the number of features in each dataset. Our network data contains 47 features in total, providing enough features for multivariate setting. This ensures that our dataset is well-suited for training TSFMs, similar to existing pre-trained datasets. In our initial experiments, we used a subset of four important features from our network dataset as mentioned in Table 2. We extended our analysis to include ten features in total, as shown in Table 8, and evaluated the performance of the benchmarked models in this multivariate setting. Table 9 shows that ARF outperforms all the other benchmarked models in this multivariate setting as well. The performance of these models is more clearly reflected in Fig. 11. We observe that ARF follows the curve/trend of the bitrate much better than other benchmarked models. All models were evaluated using their default hyper-parameters.

Table 7: Number of features in different datasets.

| Dataset | No. of features |
|---|---|
| **Network** | **47** |
| ETTh1 | 8 |
| Electricity | 322 |
| Weather | 22 |
| Traffic | 863 |

Table 8: Features used in multivariate setting.

| Feature | Description |
|---|---|
| CQI | Channel Quality Indicator |
| MCS | Modulation and Coding Scheme |
| pkt ok | Number of packets sent |
| pkt nok | Number of packets dropped |
| id_ue | Number of ue's connected in the BS |
| pusch_sinr | Noise ratio interference in the Physical Uplink Shared Channel |
| pucch_sinr | Noise ratio interference in the Physical Uplink Control Channel |
| pusch_rssi | Signal strength in the Physical Uplink Shared Channel |
| pucch_rssi | Signal strength in the Physical Uplink Control Channel |
| pucch samples | Number of PUCCH samples |

### A.5.2 FILTERED DATA EVALUATION

In this section, we evaluate the performance of the benchmarked models on mobility patterns and traffic classes that differ from those presented in Section 3.2. The raw data is filtered based on mobility patterns and traffic generated from malicious activities. In particular, we focus on the *train* mobility pattern for the *Dos-Hulk-C* traffic class. This analysis also demonstrates the potential of the dataset for transfer learning use case; by training models on one set of mobility patterns and traffic classes and evaluating them on a different set, we can assess how well knowledge learned in one context generalizes to another.

Table 10 presents the performance of ARF and TTM, evaluated using RMSE and MAE in both univariate and multivariate settings. We specifically include TTM in our analysis because of its

Table 9: Performance of benchmarked models using ten features.

| Model | Multivariate | |
|---|---|---|
| | RMSE | MAE |
| XGB | 0.0347 | 0.0234 |
| ARF | **0.0273** | **0.0155** |
| Naive | 0.0417 | 0.0239 |
| TTM (Zero-shot) | 0.0359 | 0.0230 |
| TTM (Fine-tuning) | 0.0358 | 0.0228 |
| Chronos (Zero-shot) | 0.0285 | 0.0181 |
| Chronos (Fine-tuning) | 0.0280 | 0.0176 |

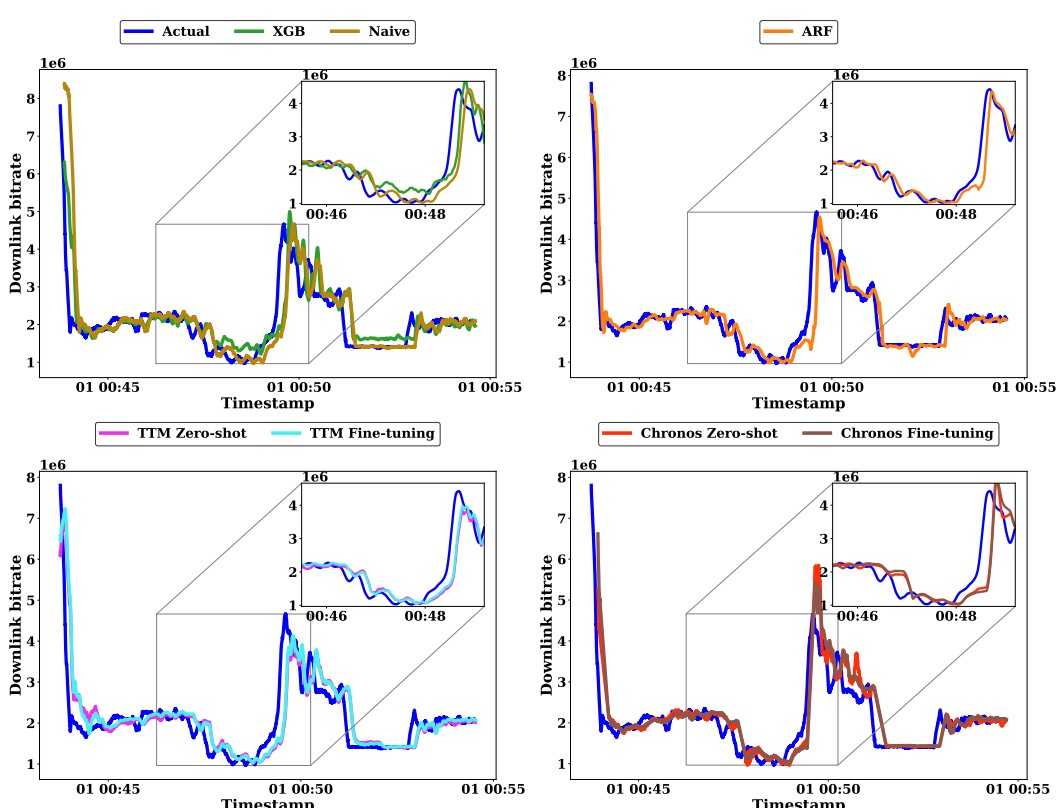

Figure 11: Actual v.s. Predicted bitrate values using ten features.

Table 10: Performance metrics of benchmarked models on a new filtered data.

| Model | Univariate | | Multivariate | |
|---|---|---|---|---|
| | RMSE | MAE | RMSE | MAE |
| XGB | 0.1440 | 0.1087 | 0.1440 | 0.1087 |
| ARF | 0.1728 | 0.1125 | **0.0968** | **0.0634** |
| Naive | 0.1309 | 0.0932 | 0.1309 | 0.0932 |
| TTM (Zero-shot) | **0.1279** | **0.0922** | 0.1279 | 0.0922 |

computational efficiency. For the filtered dataset, we observe that TTM outperforms ARF in the univariate setting. However, in the multivariate setting, ARF achieves better performance compared to the other models. Fig. 12 further illustrates how these models follow the trend of the bitrate.

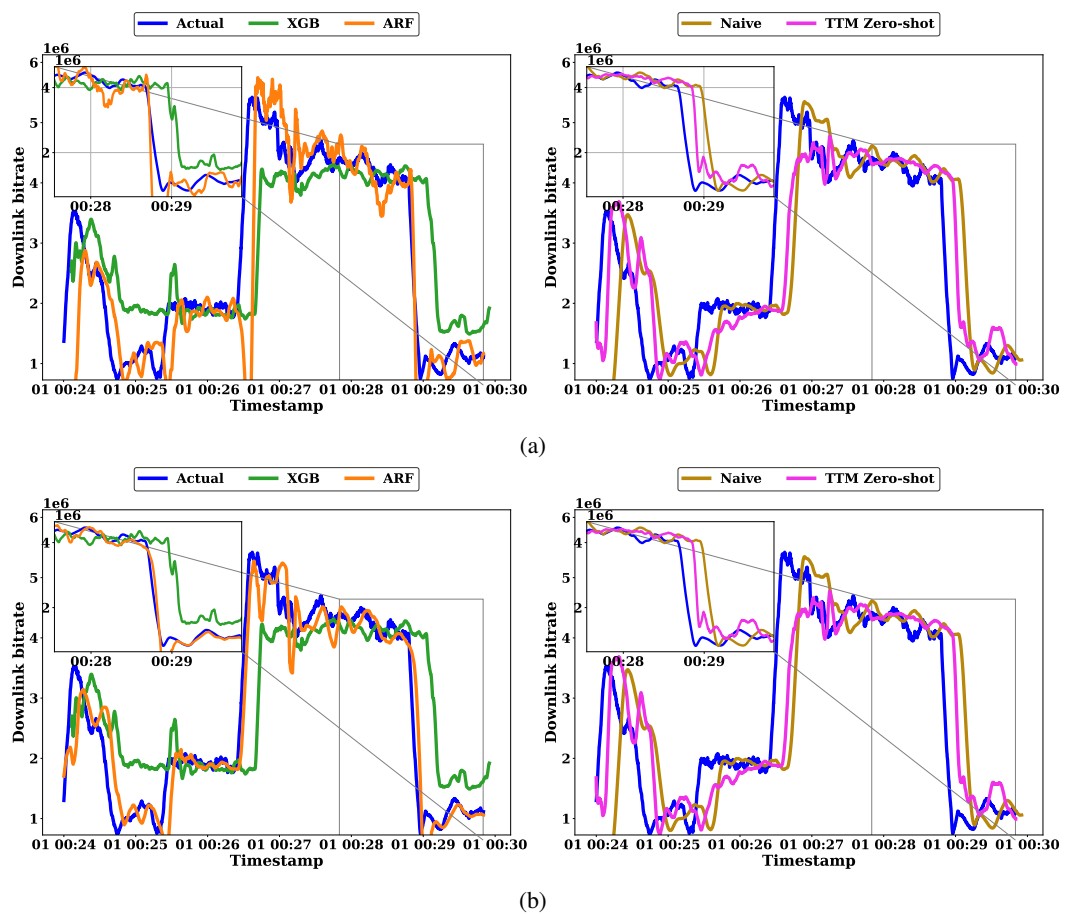

Figure 12: Actual v.s. Predicted bitrate values on a new filtered data: (a) Univariate, (b) Multivariate.

Table 11: Performance metrics of benchmarked models on a new filtered data.

| Traffic Labels | Multivariate | | | | | | | |
| | Mobility Pattern : YouTube | | | | Mobility Pattern : Portscan | | | |
| | Model | | | | | | | |
| | ARF | | TTM (Zero-shot) | | ARF | | TTM (Zero-shot) | |
| | RMSE | MAE | RMSE | MAE | RMSE | MAE | RMSE | MAE |
| Static | **0.0166** | **0.0123** | 0.0359 | 0.0230 | **0.0858** | **0.0585** | 0.1192 | 0.0904 |
| Pedestrian | **0.0457** | **0.0262** | 0.0765 | 0.0434 | **0.0819** | **0.0555** | 0.1127 | 0.0860 |
| Bus | **0.0790** | **0.0406** | 0.0805 | 0.0481 | **0.0522** | **0.0337** | 0.0945 | 0.0712 |
| Train | **0.0564** | **0.0303** | 0.0623 | 0.0443 | 0.1684 | **0.1180** | **0.1681** | 0.1308 |
| Car | **0.0404** | **0.0217** | 0.0568 | 0.0320 | **0.0819** | **0.0555** | 0.1375 | 0.1037 |

We further extend our evaluation to additional filtered subsets of our data, each representing distinct combinations of mobility patterns and traffic classes. Here, we restrict our experiments to the multivariate setting. Table 11 shows that TTM consistently has poorer performance compared to ARF in most traffic labels. These results indicate the stronger generalization of ARF in the multivariate setting.

Table 12: Hyper-parameter tuning of the TTM model in the multivariate setting.

| Learning Rate (LR) | RMSE | MAE | Fine-tune Percent | RMSE | MAE | No. of Epochs | RMSE | MAE |
|---|---|---|---|---|---|---|---|---|
| 0.01 | 0.0390 | 0.0249 | 10 | 0.0358 | 0.0227 | 50 | 0.0359 | 0.0227 |
| 0.001 | 0.0387 | 0.0247 | 15 | 0.0365 | 0.0226 | 80 | 0.0359 | 0.0227 |
| 0.00001 | 0.0359 | 0.0227 | 25 | 0.0366 | 0.0227 | 100 | 0.0359 | 0.0227 |
| 0.000001 | 0.0359 | 0.0229 | 30 | 0.0367 | 0.0226 | | | |

Table 13: Hyper-parameter tuning of Lag-Llama model.

| Context Length | RMSE | MAE | Batch Size | RMSE | MAE |
|---|---|---|---|---|---|
| 15 | 0.0350 | 0.0231 | 16 | 0.0330 | 0.0227 |
| 25 | 0.0324 | 0.0217 | 32 | 0.0314 | 0.0218 |
| 35 | 0.0327 | 0.0221 | 128 | 0.0332 | 0.0221 |

### A.5.3    TSFMs HYPER-PARAMETER TUNING

In this section, we evaluate the performance of TTM and Lag-Llama models under different hyper-parameter settings. Table 12 summarizes the results of hyper-parameter tuning for the TTM model in the multivariate setting. It presents the performance of the TTM model under various learning rates, fine-tuning percentages, and number of epochs. We first evaluated the different learning rates, observing that a learning rate of 0.00001 achieves the lowest errors. Using this optimal learning rate (0.00001), we further experimented with different fine-tuning percentages, observing that finetuning 10% of training data results in lower RMSE and while the MAE remains largely similar across all fine-tuning percentages. We also tuned the number of training epochs using this learning rate, but found that increasing the number of epochs did not significantly change either the RMSE or MAE, indicating that the performance is largely insensitive to the number of epochs beyond the default setting. TTM provides a built-in learning rate finder, which we used to determine the optimal learning rate for the main results presented in Table 4. Using the learning rate finder algorithm, we obtained an optimal learning rate of 0.0011 for the main results in Table 4, resulting in an RMSE of 0.0391 and an MAE of 0.0249. This shows that tuning the learning rate can noticeably improve the performance of the TTM model after fine-tuning. Nevertheless, ARF continues to outperform TTM.

Further, Table 13 presents the hyper-parameter tuning results for the Lag-Llama model in the uni-variate setting. We first evaluated different context lengths and observed that a context length of 25 performs better than context length 15. For the main results in Table 4, we used a context length of 5 to ensure a fair comparison with the shallow models, which operate under the same input length constraint. With this context length of 5, Lag-Llama achieves RMSE of 0.0474 and MAE of 0.0268. After selecting the best-performing context length (25), we then experimented with different batch sizes. As shown in the table, a batch size of 32 provides slightly better performance compared to the other batch sizes.

Similarly, we performed additional experiments in a univariate setting with the zero-shot Chronos-base model, which has 200M parameters, whereas the main results in Table 4 use Chronos-small with 46M parameters. Although Chronos-base achieves slightly lower errors (RMSE: 0.0294, MAE: 0.0179), the improvement over Chronos-small (RMSE: 0.0313, MAE: 0.0185) appears modest de-spite a more than fourfold increase in model size. Moreover, ARF continues to outperform Chronos in terms of RMSE, indicating that the performance gap is not solely due to model scale.

### A.6    USE OF LARGE LANGUAGE MODELS

Large language models (LLMs) have been used exclusively for the purpose of text editing.

