# OpenReview forum: "Bridging the High-Frequency Data Gap: A Millisecond-Resolution Dataset for Advancing Time Series Foundation Models"
_ICLR.cc/2026/Conference — Submitted to ICLR 2026_

### Official Review · Reviewer_4iPt · 2025-10-26

**Soundness:** 2
**Presentation:** 2
**Contribution:** 2
**Rating:** 2
**Confidence:** 4

**Summary:**

This paper investigates the important challenge of adapting TSFMs to high-frequency data, a domain that is currently underrepresented in standard pre-training benchmarks. The authors introduce a new millisecond-resolution dataset from a 5G wireless network and benchmark several TSFMs and traditional models.

**Strengths:**

1. The paper introduces a new, real-world dataset with millisecond-level resolution, addressing a significant gap since most existing benchmarks focus on low-frequency data
2. It brings the domain of wireless networks (5G) into the scope of TSFM.
3. The study effectively demonstrates that current TSFMs perform poorly on this new high-frequency dataset.
While the direction of this work is commendable, I have several major concerns that prevent me from recommending acceptance at this time.

**Weaknesses:**

While the direction of this work is commendable, I have several major concerns that prevent me from recommending acceptance at this time.
1. the paper's scope and claims seem overstated. The title and introduction promise to "Bridge the High-Frequency Data Gap" , but the contribution is a single dataset from one specific domain (5G wireless networks). It is a significant leap to assume this one dataset can adequately represent the diverse challenges of all high-frequency data, which might include high-frequency finance, industrial sensor readings, or biometrics, each with unique properties. The authors should be more precise in their claims, positioning this as a valuable benchmark for high-frequency communication network data, rather than a general solution for the entire high-frequency gap.
2. I found a jarring inconsistency in the description of the experimental methodology. In Section 4, the authors state that the multivariate setting uses "four input features". However, the corresponding Table 2, which is supposed to list these features, only provides descriptions for three (CQI, MCS, and pkt ok/nok). Besides, 3/4 features for multivariate forecasting seems too few.
3. the figure quality looks poor, which should be improved.

**Questions:**

Please answer/explain the weaknesses from me.

---

> ### Author Response · Authors · 2025-11-21
>
> Thank you for the comments.
>
> > the paper's scope and claims seem overstated. The title and introduction promise to "Bridge the High-Frequency Data Gap" , but the contribution is a single dataset from one specific domain (5G wireless networks). It is a significant leap to assume this one dataset can adequately represent the diverse challenges of all high-frequency data, which might include high-frequency finance, industrial sensor readings, or biometrics, each with unique properties. The authors should be more precise in their claims, positioning this as a valuable benchmark for high-frequency communication network data, rather than a general solution for the entire high-frequency gap.
>
> Thank you for pointing this out. We have updated the title to "Bridging the High-Frequency Data Gap: A Millisecond-Resolution Network Dataset for Advancing Time Series Foundation Models", which more accurately reflects the focus on high-frequency network data. We have also revised the Introduction to clearly position our dataset as a valuable benchmark within the 5G wireless network domain, without generalizing to all types of high-frequency data.
>
> > I found a jarring inconsistency in the description of the experimental methodology. In Section 4, the authors state that the multivariate setting uses "four input features". However, the corresponding Table 2, which is supposed to list these features, only provides descriptions for three (CQI, MCS, and pkt ok/nok). Besides, 3/4 features for multivariate forecasting seems too few.
>
> We confirm that we used four input features: CQI, MCS, pkt ok, and pkt nok for the multivariate setting. To improve clarity, we have separated pkt ok and pkt nok in Table 2. While our dataset contains 47 features in total, we selected the four most important features for predicting the downlink bitrate in the main experiments in table 4. Additionally, we conducted preliminary experiments using the top 10 features to evaluate the impact of including more inputs. We will add the results of these experiments in a new table in the Appendix of the revised draft. We will also add another table in the Appendix comparing the number of features in our network dataset with those in existing pre-trained datasets. With 47 features, our dataset provides enough features for multivariate setting, and is well-suited for training TSFMs, similar to existing pre-trained datasets.
>
> *Performance of benchmarked models with ten features.*
>
> | **Model**              | **RMSE**      | **MAE**       |
> |------------------------|---------------|---------------|
> | XGB                   | 0.0347        | 0.0234        |
> | ARF                   | **0.0273**    | **0.0155**    |
> | Naive                 | 0.0417        | 0.0239        |
> | TTM (Zero-shot)       | 0.0359        | 0.0230        |
> | TTM (Fine-tuning)     | 0.0358        | 0.0228        |
> | Chronos (Zero-shot)   | 0.0285        | 0.0181        |
>
>
> *Number of features in different datasets.*
>
> | **Dataset**     | **No. of features** |
> |-----------------|----------------------|
> | Network         | 47                  |
> | ETTh1           | 8                   |
> | Electricity     | 322                 |
> | Weather         | 22                  |
> | Traffic         | 863                 |
>
> > the figure quality looks poor, which should be improved.
>
> We have improved the resolution and clarity of all figures to ensure they are easily readable and visually clear in the revised draft.
>
> We hope this reply answers your questions satisfactorily. We are available for discussion of any further questions or doubts you may have regarding our work.

---

> > ### Author Response · Authors · 2025-12-02
> >
> > Please find below the updated changes in the revised paper.
> >
> > * **W2** We conducted additional experiments using the top 10 features to evaluate the impact of including more inputs. We have added the results of these experiments in a new table in the Appendix of the revised draft.
> >
> >
> > *Performance of benchmarked models with ten features*
> >
> > | **Model**               | **RMSE**      | **MAE**       |
> > |-------------------------|---------------|---------------|
> > | XGB                    | 0.0347        | 0.0234        |
> > | ARF                    | **0.0273**    | **0.0155**    |
> > | Naive                  | 0.0417        | 0.0239        |
> > | TTM (Zero-shot)        | 0.0359        | 0.0230        |
> > | TTM (Fine-tuning)      | 0.0358        | 0.0228        |
> > | Chronos (Zero-shot)    | 0.0285        | 0.0181        |
> > | Chronos (Fine-tuning)  | 0.0280        | 0.0176        |

---

### Official Review · Reviewer_NhZy · 2025-10-28

**Soundness:** 2
**Presentation:** 3
**Contribution:** 2
**Rating:** 4
**Confidence:** 4

**Summary:**

This paper introduces and analyzes a new millisecond-resolution time series dataset derived from a real-world operational 5G wireless network environment, focusing on wireless and traffic measurement data. Unlike existing large-scale time series foundation model (TSFM) benchmarks, which are dominated by low-frequency domains and traditional sources such as finance, energy, and general sensor data, this dataset captures high-frequency wireless conditions and supports short-term forecasting tasks, with horizons spanning from 100 milliseconds to nearly 10 seconds. The authors perform a comprehensive empirical evaluation using several shallow learning baselines (RF, XGBoost, ARF, Naive) and modern TSFMs (TTM, Chronos, Lag-Llama), assessing their performance on both univariate and multivariate tasks, and provide detailed analysis of the unique data characteristics involved. The key finding is that TSFMs, even after fine-tuning, perform poorly on this data compared to adaptive shallow learners, highlighting fundamental gaps in TSFM generalization to high-frequency, irregular environments.

**Strengths:**

**S1** The paper addresses a major gap in the current TSFM landscape by introducing a uniquely high-frequency (millisecond-level) real-world dataset from the wireless networking domain, as clearly contrasted in Figure 1 and Figure 2. The data provides a new benchmark for both model development and evaluation.

**S2** Strong empirical analysis is presented demonstrating, with quantitative rigor (see Table 4, Table 5), that state-of-the-art TSFMs underperform relative to adaptive shallow learners, particularly ARF, in the irregular and bursty environment of 5G wireless bitrate prediction.

**S3** Figure 4 and the associated discussion show thorough exploration of the dataset’s temporal and statistical characteristics—decomposing trend, seasonality, residuals, stationarity, heavy-tailed behavior, and autocorrelation structure—a critical diagnostic for understanding challenges in model generalization.

**Weaknesses:**

**W1** Empirical Scope – Dataset Subset and Generalizability: While the raw dataset is described as highly diverse (various mobility and traffic patterns, including adversarial traffic), all primary experiments focus on a filtered subset (static mobility, benign video traffic). Results may thus lack generality for the broader dataset or for "active" network regimes expected in practice. See Section 3.2 and Section 4.3 for specifics. Although an ablation is provided in Appendix A.3, it is minor relative to the main paper’s claims and only briefly covers a single alternative pattern (train mobility, DoS-Hulk-C traffic class).

**W2** Limited TSFM Fine-tuning and Adaptation: The TSFM models are mainly used in zero-shot and vanilla fine-tuning modes, with little attention given to recent adaptation/transfer techniques (e.g., domain adaptation, LoRA, sophisticated feature engineering, deep calibration, or per-task tuning). While Limitations (Section 5) mention this, the omission is significant, especially since ARF’s advantage may partly stem from online adaptation absent in the TSFM attempts.

**W3** Baselines: Shallow models use modest hyperparameter tuning, and TSFM implementations are described as "default." More systematic and rigorous HPO or ensembling (which is routine for robust time series baselines) could potentially close some of the reported performance gap, as acknowledged in Section 5, raising questions about the finality of the reported results.

**W4** Metrics and Fairness of Comparison: Although considerable care goes into aligning horizon evaluation, there are differences in how the models exploit input features, prediction structure, and context. For example, Chronos is omitted from the multivariate analysis, potentially limiting the claim of absolute TSFM underperformance—the models may simply not be fully leveraged in this regime.

**Questions:**

**Q1** Can the authors provide a more thorough examination of why adaptive shallow methods (e.g., ARF) outperform TSFMs? Is the primary issue lack of high-frequency data in pre-training, or are the architectural/modeling choices themselves inadequate for volatility and concept drift?

**Q2** What experimental results, if any, are available for larger-scale TSFMs or multiscale/transfer models? Could these models narrow the observed gap with more targeted fine-tuning or adaptation?

---

> ### Author Response · Authors · 2025-11-21
>
> > Can the authors provide a more thorough examination of why adaptive shallow methods (e.g., ARF) outperform TSFMs? Is the primary issue lack of high-frequency data in pre-training, or are the architectural/modeling choices themselves inadequate for volatility and concept drift?
>
> Our analysis suggests that ARF outperforms TSFMs because it handles bursts of volatility and concept drift more effectively, which are dominant in our high-frequency dataset. Even with larger models like Chronos-base model or hyperparameter tuning of TTM or Lag-Llama, TSFMs do not match performance of ARF, indicating that both the pre-training data and the model design contribute to their limitations. To further validate, we conducted an experimental analysis to evaluate how TSFMs performance changes under different sampling frequencies, focusing initially on TTM in the zero-shot setting. Our initial results indicate that as the temporal resolution increases, the performance of TTM improves. These findings highlight the value of including high-frequency data like ours in the pre-training datasets. We will add the results for other TSFMs, as well as for TTM fine-tuning, in the revised draft.
>
> *Performance of TTM Zero-shot forecasting with the increasing temporal resolution.*
> | **Temporal Resolution** | **Prediction Horizon** | **RMSE**  | **MAE**   |
> |--------------------------|-------------------------|-----------|-----------|
> | 200 ms                   | 48                      | 0.0788    | 0.0472    |
> | 500 ms                   | 20                      | 0.0997    | 0.0598    |
> | 1000 ms                      | 10                      | 0.1117    | 0.0683    |
> | 2000 ms                      | 5                       | 0.1514    | 0.0941    |
> | 3000 ms                     | 4                       | 0.1765    | 0.1096    |
>
> > What experimental results, if any, are available for larger-scale TSFMs or multiscale/transfer models? Could these models narrow the observed gap with more targeted fine-tuning or adaptation?
>
> We conducted additional experiments using the zero-shot Chronos-base model, which has 200M parameters, while the main results used Chronos-small with 46M parameters. Although Chronos-base achieves slightly lower errors; RMSE:0.0294 and MAE:0.0179, the improvement over Chronos-small; RMSE:0.0313, MAE:0.0185 appears modest, despite a more than fourfold increase in model size. Moreover, ARF continues to outperform Chronos, indicating that the performance gap is not solely due to model scale.
>
> We would like to emphasize that the code and the dataset will be open-sourced after the reviews for the community and our own research and ablation studies to continue on this dataset and the relevance for TSFMs.
>
> We expect this reply clears up your doubts regarding our paper. In any case, we are available for further discussions.

---

> > ### Author Response · Authors · 2025-12-03
> >
> > * **Q1** Our network data is sampled at very high frequency, which makes short-term fluctuations, bursts of volatility, and clustered anomalies the dominant patterns. This is supported by SNR analysis: the time series is dominated by short-term periodic structures (high SNR in periods 2-20), while medium-term cycles exist but are weaker, and long-term seasonality is essentially absent (SNR nears to zero and even negative beyond period 600). These characteristics explain why models that capture short-term dynamics perform better in our experiments. To further validate, we conducted an experimental analysis to evaluate how the performance of ARF and TSFMs changes under different sampling frequencies in a multivariate setting, performing fine-tuning only for TTM because of its computational efficiency. The prediction horizon is fixed at 9.6 seconds for all temporal resolutions. Specifically, we evaluate these models on a newly filtered data; \textit{pedestrian} mobility pattern for the \textbf{video streaming} traffic class, to highlight that the characteristics of our dataset differ from those of the pre-trained datasets. Our results indicate that increasing the temporal resolution does not improve the performance of TSFMs. In contrast, ARF consistently outperforms TSFMs at each resolution, as higher temporal resolution reduces noise and improves its predictions. This indicates that TSFMs perform poorly not only because of temporal resolution (i.e., high frequency), but also due to the inherent characteristics of our data. These findings highlight the value of including high-frequency data like ours in the pre-training datasets.
> >
> >
> > *Performance metrics of benchmarked models with the increasing temporal resolution*
> >
> > | **Temporal Resolution** | **Prediction Horizon** | **ARF (RMSE / MAE)** | **TTM Zero-shot (RMSE / MAE)** | **TTM Fine-tuning (RMSE / MAE)** | **Chronos Zero-shot (RMSE / MAE)** |
> > |--------------------------|-------------------------|-----------------------|---------------------------------|-----------------------------------|-------------------------------------|
> > | 100 ms                  | 96                      | **0.0457 / 0.0262**  | 0.0765 / 0.0434                | 0.0743 / 0.0421                  | 0.0622 / 0.0338                    |
> > | 200 ms                  | 48                      | **0.0471 / 0.0267**  | 0.0870 / 0.0499                | 0.0880 / 0.0496                  | 0.0740 / 0.0389                    |
> > | 500 ms                  | 20                      | **0.0398 / 0.0218**  | 0.0855 / 0.0490                | 0.0894 / 0.0542                  | 0.0711 / 0.0372                    |
> > | 1000 ms                 | 10                      | **0.0297 / 0.0176**  | 0.0856 / 0.0500                | 0.0856 / 0.0500                  | 0.0580 / 0.0326                    |
> > | 2000 ms                 | 5                       | **0.0289 / 0.0169**  | 0.0880 / 0.0527                | 0.0915 / 0.0584                  | 0.0671 / 0.0354                    |
> > | 3000 ms                 | 4                       | **0.0289 / 0.0185**  | 0.1049 / 0.0618                | 0.1061 / 0.0638                  | 0.0860 / 0.0443                    |

---

> ### Comment · Reviewer_NhZy · 2025-11-22
>
> Thanks for the response. After i read your rebuttal for me and other reviews, i find that you update the results of Chronos (fine-tune) to RMSE:0.0282 and MAE:0.0178. Will these updates affect the conclusions of this paper? I still have concerns about W2. If more fine-tuning techniques are applied, will the results of TSFMs outperform shallow models? I think even the original result of Chronos (zero-shot) with MAE: 0.0185 is still better than the marked ARF.
>
> And there still exists some lightweight TSFMs, such as LightGTS, TTM, UniTS, with only 1M--4M parameters. You may consider them and finetune them to validate your conclusions. If the authors better address W2, i will raise my rates to support acceptance.

---

> > ### Author Response · Authors · 2025-12-02
> >
> > Please find below the updated changes in the revised paper.
> >
> > * **W2** We performed basic fine-tuning for TTM to assess its performance. We explore two different fine-tuning strategies: \(i) Head-only fine-tuning , where we freeze the entire backbone and decoder and only train the final prediction head, (ii) Adapter-based fine-tuning, where we incorporate lightweight MLP adapter modules inside the mixer blocks while keeping the original TTM weights frozen. Recent works on fine-tuning TSFMs (Tomar, Shivani, et al. "AT4TS: Autotune for Time Series Foundation Models." Transactions on Machine Learning Research.) has shown that even widely used PEFT methods like LoRA do not consistently improve the performance of TSFMs. Our findings aligns with this observation; even though both the fine-tuning strategies are architecturally compatible with TTM, their performance is worse as compared to default TTM fine-tuning approach.  We have added these results in the revised draft.
> >
> >
> > *Performance Metrics for Different Fine-Tuning Strategies for TTM*
> >
> > | **Fine-tuning Strategy**       | **RMSE**  | **MAE**   |
> > |--------------------------------|-----------|-----------|
> > | Head-only fine-tuning          | 0.0413    | 0.0270    |
> > | Adapter-based fine-tuning      | 0.0522    | 0.0334    |

---

### Official Review · Reviewer_uVpw · 2025-10-29

**Soundness:** 1
**Presentation:** 2
**Contribution:** 1
**Rating:** 4
**Confidence:** 4

**Summary:**

This paper introduces a new, real‑world, millisecond‑resolution time‑series dataset collected from a 5G Open RAN deployment with a focus on short-horizon forecasting. The authors benchmark traditional shallow models (RF, XGB, ARF) and several TSFMs (TTM, Chronos‑bolt‑small, Lag‑Llama) on univariate and multivariate forecasting of downlink bitrate, showing that an online/streaming approach outperforms both static shallow models and current TSFMs on this high‑frequency, spiky, non‑stationary data. The paper argues that existing TSFMs generalize poorly to this regime and calls for incorporating high‑frequency domains like wireless networking in pretraining corpora.

**Strengths:**

1. The figures contrasting timescales and domains make a strong case that current pretraining corpora underrepresent millisecond‑level data and the wireless domain
2. Measurements are from an operational O‑RAN (with near‑RT RIC) using USRPs and diverse mobility/traffic profiles, which increases ecological validity over synthetic lab traces.
3. The paper analyzes non‑stationarity, heavy tails, weak or absent seasonality and clustered extremes using STL/rolling stats/Q‑Q/SNR/ACF explaining why generic TSFMs struggle.

**Weaknesses:**

1. Chronos is evaluated only univariately (Section 4.1), while shallow models in multivariate mode leverage exogenous features
2. For Chronos, zero‑shot and fine‑tuned metrics are identical to four decimals (0.0313 MAE 0.0185). This is suspicious and suggests fine‑tuning may not have actually changed the model or was evaluated incorrectly.
3. For TTM, fine‑tuning is worse than zero‑shot in the multivariate setting (Table 4), which deserves diagnosis beyond claiming suboptimal.
4. Section 5 acknowledges minimal HPO for RF/XGB and default TSFM configs. Given the strong claim about “TSFMs perform poorly,” a modest, targeted sweep (e.g., context length, learning rate, adapter size) and a couple of online baselines (e.g., simple online linear/Kalman filters) would increase credibility
5. It’s unclear how train/val/test splitting is done (“80:20” is given, but is the split strictly temporal and by UE?). Seed control and repetition are not discussed; many ± values in Table 4 are 0.0000, suggesting a single deterministic run or missing variance estimation.

**Questions:**

Please check the weaknesses above

---

> ### Author Response · Authors · 2025-11-21
>
> Thank you for your comments.
> > Chronos is evaluated only univariately (Section 4.1), while shallow models in multivariate mode leverage exogenous features.
>
> Chronos is inherently a univariate forecasting model, so to enable multivariate evaluation we followed the approach provided by AutoGluon-TimeSeries (AG-TS) using covariate regressors. A covariate regressor predicts the target from covariates and static features; its output is subtracted, and a univariate model forecasts the residuals. Using this setup, we conducted an initial study for zero-shot multivariate experiments and obtained RMSE: 0.0273 and MAE: 0.0181.  However, we note that ARF still outperforms Chronos in this multivariate setting, reinforcing our earlier analysis. We will extend these experiments to additional seeds and also evaluate Chronos fine-tuning in the multivariate setting, and we will update Table 4 accordingly in the revised draft.
>
> > For Chronos, zero‑shot and fine‑tuned metrics are identical to four decimals (0.0313 MAE 0.0185). This is suspicious and suggests fine‑tuning may not have actually changed the model or was evaluated incorrectly.
>
> Thank you for highlighting this issue. The identical zero-shot and fine-tuned metrics for Chronos indeed indicated a problem. Upon re-examining the code, we identified and corrected an error in the fine-tuning method. We fixed the issue, and reran the experiments and obtained updated results of RMSE:0.0282 and MAE:0.0178. The corrected fine-tuning code has been executed for a single seed (42). We will rerun the fine-tuning experiments with the remaining two seeds and update Table 4 accordingly in the revised draft. However, ARF still outperforms Chronos even after fine-tuning the model.
>
> > For TTM, fine‑tuning is worse than zero‑shot in the multivariate setting (Table 4), which deserves diagnosis beyond claiming suboptimal.
>
> We conducted additional hyper-parameter tuning to better understand this behavior. We found that the results presented in Table 4 were primarily driven by the choice of learning-rate during fine-tuning. TTM includes a built-in learning-rate finder, which we applied to our main experiments. Using the learning rate finder algorithm, we obtained an optimal learning rate of 0.00109749876 for the main results in Table 4, resulting in RMSE of 0.0391 and MAE of 0.0249. To further validate this, we performed additional experiments across a wider range of learning rates, including 0.1, 0.01, 0.00001, and 0.000001. Among these, the learning rate of 0.00001 achieves the lowest errors, resulting in RMSE of 0.0359 and MAE of 0.0227. However, ARF still outperforms TTM even after the hyper-parameter tuning.  We have added a hyper-parameter tuning section of TSFMs in Appendix in the revised version. The code and the dataset will be open-sourced after the reviews for the community and our own research and ablation studies to continue on this dataset and the relevance for TSFMs.

---

> ### Author Response · Authors · 2025-11-21
>
> > Section 5 acknowledges minimal HPO for RF/XGB and default TSFM configs. Given the strong claim about “TSFMs perform poorly,” a modest, targeted sweep (e.g., context length, learning rate, adapter size) and a couple of online baselines (e.g., simple online linear/Kalman filters) would increase credibility.
>
> We conducted preliminary hyper-parameter tuning for the TSFM models, specifically for TTM and Lag-Llama. For TTM, we performed tuning over learning rates and different fine-tune percentage. For Lag-Llama, we performed a tuning over context length and batch size. Even after hyper-parameter tuning, ARF still outperforms both TTM and Lag-Llama. We will include these results in the Appendix and add the suggested online baselines to our experiments in the revised draft.
>
> *Hyper-parameter tuning of the TTM model in the multivariate setting.*
>
> | **Learning Rate (LR)** | **RMSE**  | **MAE**   | **Fine-tune Percent** | **RMSE**  | **MAE**   |
> |-------------------------|-----------|-----------|------------------------|-----------|-----------|
> | 0.01                   | 0.0390    | 0.0249    | 10                     | 0.0358    | 0.0227    |
> | 0.001                  | 0.0387    | 0.0247    | 15                     | 0.0365    | 0.0226    |
> | 0.00001                | 0.0359    | 0.0227    |                        |           |           |
>
>
> *Hyper-parameter tuning of Lag-Llama model.*
>
> | **Context Length** | **RMSE**  | **MAE**   | **Batch Size** | **RMSE**  | **MAE**   |
> |---------------------|-----------|-----------|-----------------|-----------|-----------|
> | 15                  | 0.0350    | 0.0231    | 32              | 0.0314    | 0.0218    |
> | 25                  | 0.0324    | 0.0217    | 128             | 0.0332    | 0.0221    |
>
>
> > It’s unclear how train/val/test splitting is done (“80:20” is given, but is the split strictly temporal and by UE?). Seed control and repetition are not discussed; many ± values in Table 4 are 0.0000, suggesting a single deterministic run or missing variance estimation.
>
> Thank you for this valuable comment. We have clarified the data splitting procedure and seed handling which is added in the rebuttal revision. Specifically, the dataset was divided into 80\% training and 20\% testing while strictly preserving temporal order. Because the data for each user are sequential and not mixed, this split naturally keeps the sequence of each user intact, preventing data leakage from future observations into training. Regarding seed control and repetition, the XGB and RF models were trained multiple times using different seed values (42, 99, 123) to estimate variability. However, all repeated runs produced identical RMSE and MAE values, resulting in a standard deviation of 0.0000. We found that the results were deterministic because the models used no stochastic operations (no subsampling or feature sampling or bootstraping), so every run produced exactly the same trees and identical predictions. We will update Table 4 after making the necessary changes. For the fine-tuning experiments involving TTM and Chronos, we also identified and corrected issues in the original code, and the updated results will be included in the revised draft.
>
> We hope the details and clarifications given in this rebuttal aids in communicating the significance and originality of the paper. In any case, we are available for further discussions.

---

> > ### Author Response · Authors · 2025-12-02
> >
> > Please find below the updated changes in the revised paper.
> > * **W1**: Chronos is inherently a univariate forecasting model. To enable multivariate evaluation, we followed the approach provided by AutoGluon-TimeSeries (AG-TS) using covariate regressors. Using this setup, we initially conducted zero-shot multivariate experiments, obtaining RMSE = 0.0273 and MAE = 0.0181. We have now also performed fine-tuning for Chronos in the multivariate setting, resulting in improved performance (RMSE = 0.0253, MAE = 0.0176). Despite this improvement, ARF still outperforms Chronos, reinforcing our earlier analysis. We have updated the Table 4 in the revised draft to reflect these new results.
> >
> > * **W2**: We fixed the issue in the fine-tuning code, and reran the experiments for three different seeds and obtained updated results of RMSE:0.0282 and MAE:0.0178. We have updated Table 4 accordingly in the revised draft. While Chronos performs well in terms of MAE in the univariate setting, ARF consistently outperforms all the models in the multivariate setting.
> >
> > * **W3**: We have also added the suggested online baseline, a simple incremental linear regression model (Online Linear Regression), and validated that that ARF still outperforms it. We have updated Table 4 with these results as well.
> >
> > * **W4**: We conducted hyper-parameter tuning for the TSFM models, specifically for TTM and Lag-Llama. For TTM, we performed tuning over learning rates, different fine-tune percentage, and number of epochs. For Lag-Llama, we performed a tuning over context length and batch size. Even after hyper-parameter tuning, ARF still outperforms both TTM and Lag-Llama. We have added these results to the Appendix in the revised draft.
> >
> > *Hyper-parameter tuning of the TTM model in the multivariate setting*
> >
> > | **Learning Rate (LR)** | **RMSE** | **MAE** | **Fine-tune Percent** | **RMSE** | **MAE** | **No. of Epochs** | **RMSE** | **MAE** |
> > |-------------------------|----------|---------|------------------------|----------|---------|--------------------|----------|---------|
> > | 0.01                   | 0.0390   | 0.0249  | 10                     | 0.0358   | 0.0227  | 50                 | 0.0359   | 0.0227  |
> > | 0.001                  | 0.0387   | 0.0247  | 15                     | 0.0365   | 0.0226  | 80                 | 0.0359   | 0.0227  |
> > | 0.00001                | 0.0359   | 0.0227  | 25                     | 0.0366   | 0.0227  | 100                | 0.0359   | 0.0227  |
> > | 0.000001               | 0.0359   | 0.0229  | 30                     | 0.0367   | 0.0226  |                    |          |         |
> >
> >
> > *Hyper-parameter tuning of Lag-Llama model*
> >
> > | **Context Length** | **RMSE** | **MAE** | **Batch Size** | **RMSE** | **MAE** |
> > |---------------------|----------|---------|-----------------|----------|---------|
> > | 15                  | 0.0350   | 0.0231  | 16              | 0.0330   | 0.0227  |
> > | 25                  | 0.0324   | 0.0217  | 32              | 0.0314   | 0.0218  |
> > | 35                  | 0.0327   | 0.0221  | 128             | 0.0332   | 0.0221  |
> >
> > * **W5** We have updated Table 4 after making necessary changes in the code regarding seed control and repetition in the revised version.

---

### Official Review · Reviewer_qbv6 · 2025-10-31

**Soundness:** 2
**Presentation:** 2
**Contribution:** 3
**Rating:** 4
**Confidence:** 4

**Summary:**

This paper introduces a new high-frequency (millisecond-resolution) time series dataset collected from a real-world 5G Open RAN deployment, targeting a significant gap in existing time series foundation model (TSFM) resources. The dataset extends temporal coverage to very fine granularity and offers a benchmark scenario for short-term forecasting tasks relevant to wireless communication networks. The authors provide a detailed characterization of the dataset, positioning it against standard low-frequency TSFM benchmarks, and perform comparative evaluation of traditional shallow models and several prominent TSFMs. Results indicate substantial challenges for current TSFMs when applied to high-frequency wireless data, both in zero-shot and fine-tuned setups. The work highlights the need for broader data diversity—including high-frequency wireless contexts—in future TSFM research.

**Strengths:**

1. The presented dataset addresses a clear and important gap in the current TSFM ecosystem, focusing on an underrepresented, high-frequency (millisecond-resolution) regime that existing benchmarks neglect (see Figures 1, 2, and 3). The clear presentation of this gap is a notable strength.
2. The data collection methodology is well described, leveraging a fully operational 5G O-RAN setup with diverse traffic types and mobility patterns. This realism and diversity enhance potential research utility for modeling real-world dynamics.
3. Explicit quantitative and qualitative analyses (Table 4, Figure 5) convincingly show how existing TSFMs struggle with this form of data, particularly emphasizing weaknesses in handling abrupt shifts, volatility, and concept drift compared to dynamic shallow learners (e.g., ARF).

**Weaknesses:**

1. The analysis is largely focused on a filtered subset (static mobility + video streaming) of the wireless dataset for primary benchmarking (Section 3.2, Section 4). The multivariate scenario has only four features, and the diversity of forecasting settings is quite restricted. This sharply limits the generality of conclusions and prevents deeper insight into the strengths and limitations of models under wider operational variations (mobility, adversarial traffic).
2. The fine-tuning protocol for TSFMs is only minimally described and does not explore more advanced adaptation mechanisms such as LoRA or domain-aware feature engineering, despite suggesting this in the limitations. Results in Table 4 may therefore understate TSFM potential.
3. A critical flaw in the paper's experimental design is the incomplete evaluation of Time Series Foundation Models (TSFMs) in the multivariate forecasting scenario. The authors introduce a multivariate dataset, making multivariate prediction a core and essential application. However, in Table 4, the performance results for TSFMs in the multivariate setting are provided only for TTM, while the results for Chronos and Lag-Llama are conspicuously absent.

**Questions:**

1. Given the observed poor performance of TSFMs, to what extent do pretraining corpus choices (domain/frequency/resolution) dominate over architectural differences? Could the authors provide more nuanced ablation on TSFM pretraining regimes?
2. Were advanced TSFM fine-tuning strategies (e.g., LoRA, domain adaptation, feature normalization) attempted, and if so, with what result? If not, does the team plan to pursue these in future benchmarking?
3. Considering the analysis in Figures 6–10 of the appendix: Can the authors explicitly discuss what feature types (seasonality, volatility, autocorrelation) drive performance differences between models?

---

> ### Author Response · Authors · 2025-11-21
>
> Thank you for your comments.
>
> > Given the observed poor performance of TSFMs, to what extent do pretraining corpus choices (domain/frequency/resolution) dominate over architectural differences? Could the authors provide more nuanced ablation on TSFM pretraining regimes?
>
> We agree that understanding how pretraining corpus characteristics such as domain, frequency, and resolution dominate over architectural differences is an important direction for deeper analysis. However, such an experimental analysis requires a study of its own and is beyond the scope of our current work. Our main goal in this paper is to evaluate and fine-tune existing TSFMs on a new domain, i.e., wireless networks and new frequency (100 ms). Benchmarking TSFMs in this frequency and domain appears to be limited, and this work aims to provide an initial evaluation in this context. The motivation for this novel dataset and relevance for TSFMs stems from the Chronos and TTM papers where the importance of resolution and domain diversity was studied to be useful. The ablation study from the TTM paper in Tiny Time Mixers (TTMs) paper  (Ekambaram, Vijay, et al. "Tiny Time Mixers (TTMs): Fast Pre-trained Models for Enhanced Zero/Few-Shot Forecasting of Multivariate Time Series"  arxiv 2024) demonstrated the importance of resolution diversity. Quoting from the paper in Section 4.7 "These experiments highlight that while the quantity of pre-training data is significant, the quality of the data, especially in terms of resolution diversity and coverage, is even more crucial for improving the model performance".
>
> > Were advanced TSFM fine-tuning strategies (e.g., LoRA, domain adaptation, feature normalization) attempted, and if so, with what result? If not, does the team plan to pursue these in future benchmarking?
>
> Following to our response in your comment 1), we did not experiment with advanced fine-tuning strategies such as LoRA, domain adaptation, or feature normalization in the current version of the work. Our objective was on evaluating fine-tuning performance of TSFMs using the default configurations. However, as the reviewer suggested, in our future work, we plan to explore these strategies, as in the paper: AT4TS : Autotune for Time Series Foundation Models (Tomar, Shivani, et al. "AT4TS: Autotune for Time Series Foundation Models." Transactions on Machine Learning Research.), which proposes an automated fine-tuning framework that systematically explores tuneable hyperparameter configurations for fine-tuning pretrained TSFMs. This paper provides the initial study for TSFMs. The code and the dataset will be open-sourced after the reviews for the community and our own research to continue on this dataset and the relevance for TSFMs.
>
> > Considering the analysis in Figures 6–10 of the appendix: Can the authors explicitly discuss what feature types (seasonality, volatility, autocorrelation) drive performance differences between models?
>
> Our network data is sampled at very high frequency, which makes short-term fluctuations, bursts of volatility, and clustered anomalies the dominant patterns. This is supported by SNR analysis: the time series is dominated by short-term periodic structures (high SNR in periods 2-20), while medium-term cycles exist but are weaker, and long-term seasonality is essentially absent (SNR nears to zero and even negative beyond period 600). These characteristics explain why models that capture short-term dynamics perform better in our experiments. To further validate, we conducted an experimental analysis to evaluate how TSFMs performance changes under different sampling frequencies, focusing initially on TTM in the zero-shot setting. Our initial results indicate that as the temporal resolution increases, the performance of TTM improves. These findings highlight the value of including high-frequency data like ours in the pre-training datasets. We will add the results for other TSFMs, as well as for TTM fine-tuning, in the revised draft.
>
> *Performance of TTM Zero-shot forecasting with the increasing temporal resolution.*
> | **Temporal Resolution** | **Prediction Horizon** | **RMSE**  | **MAE**   |
> |--------------------------|-------------------------|-----------|-----------|
> | 200 ms                   | 48                      | 0.0788    | 0.0472    |
> | 500 ms                   | 20                      | 0.0997    | 0.0598    |
> | 1000 ms                      | 10                      | 0.1117    | 0.0683    |
> | 2000 ms                      | 5                       | 0.1514    | 0.0941    |
> | 3000 ms                      | 4                       | 0.1765    | 0.1096    |
>
> We hope this reply answers your questions satisfactorily. We are available for further discussion.

---

> ### Author Response · Authors · 2025-12-02
>
> Please find below the updated changes in the revised paper.
> * **Q2**: We have now performed basic fine-tuning for TTM to assess its performance. We explore two different fine-tuning strategies: (i) Head-only fine-tuning , where we freeze the entire backbone and decoder and only train the final prediction head, (ii) Adapter-based fine-tuning, where we incorporate lightweight MLP adapter modules inside the mixer blocks while keeping the original TTM weights frozen. Recent works on fine-tuning TSFMs (Tomar, Shivani, et al. "AT4TS: Autotune for Time Series Foundation Models." Transactions on Machine Learning Research.) has shown that even widely used PEFT methods like LoRA do not consistently improve the performance of TSFMs. Our findings aligns with this observation; even though both the fine-tuning strategies are architecturally compatible with TTM, their performance is worse as compared to default TTM fine-tuning approach.  We have added these results in the revised draft.
>
> *Performance Metrics for Different Fine-Tuning Strategies for TTM*
>
> | **Fine-tuning Strategy**      | **RMSE**  | **MAE**   |
> |--------------------------------|-----------|-----------|
> | Head-only fine-tuning          | 0.0413    | 0.0270    |
> | Adapter-based fine-tuning      | 0.0522    | 0.0334    |
>
> * **Q3**: Our network data is sampled at very high frequency, which makes short-term fluctuations, bursts of volatility, and clustered anomalies the dominant patterns. This is supported by SNR analysis: the time series is dominated by short-term periodic structures (high SNR in periods 2-20), while medium-term cycles exist but are weaker, and long-term seasonality is essentially absent (SNR nears to zero and even negative beyond period 600). These characteristics explain why models that capture short-term dynamics perform better in our experiments. To further validate, we conducted an experimental analysis to evaluate how the performance of ARF and TSFMs changes under different sampling frequencies in a multivariate setting, performing fine-tuning only for TTM because of its computational efficiency. The prediction horizon is fixed at 9.6 seconds for all temporal resolutions. Specifically, we evaluate these models on a newly filtered data; 'pedestrian' mobility pattern for the 'video streaming' traffic class, to highlight that the characteristics of our dataset differ from those of the pre-trained datasets. Our results indicate that increasing the temporal resolution does not improve the performance of TSFMs. In contrast, ARF consistently outperforms TSFMs at each resolution, as higher temporal resolution reduces noise and improves its predictions. This indicates that TSFMs perform poorly not only because of temporal resolution (i.e., high frequency), but also due to the inherent characteristics of our data. These findings highlight the value of including high-frequency data like ours in the pre-training datasets.
>
>
> *Performance metrics of benchmarked models with the increasing temporal resolution*
>
> | **Temporal Resolution** | **Prediction Horizon** | **ARF (RMSE / MAE)** | **TTM Zero-shot (RMSE / MAE)** | **TTM Fine-tuning (RMSE / MAE)** | **Chronos Zero-shot (RMSE / MAE)** |
> |--------------------------|-------------------------|-----------------------|---------------------------------|-----------------------------------|-------------------------------------|
> | 100 ms                  | 96                      | **0.0457 / 0.0262**  | 0.0765 / 0.0434                | 0.0743 / 0.0421                  | 0.0622 / 0.0338                    |
> | 200 ms                  | 48                      | **0.0471 / 0.0267**  | 0.0870 / 0.0499                | 0.0880 / 0.0496                  | 0.0740 / 0.0389                    |
> | 500 ms                  | 20                      | **0.0398 / 0.0218**  | 0.0855 / 0.0490                | 0.0894 / 0.0542                  | 0.0711 / 0.0372                    |
> | 1000 ms                 | 10                      | **0.0297 / 0.0176**  | 0.0856 / 0.0500                | 0.0856 / 0.0500                  | 0.0580 / 0.0326                    |
> | 2000 ms                 | 5                       | **0.0289 / 0.0169**  | 0.0880 / 0.0527                | 0.0915 / 0.0584                  | 0.0671 / 0.0354                    |
> | 3000 ms                 | 4                       | **0.0289 / 0.0185**  | 0.1049 / 0.0618                | 0.1061 / 0.0638                  | 0.0860 / 0.0443                    |

---

### Author Response · Authors · 2025-12-02

We sincerely thank the reviewers and meta-reviewer for their time, valuable comments, and feedback. We appreciate that all the reviewers find immense value in our work, particularly its contribution to addressing major gap in the current TSFM landscape by introducing a unique, high-frequency (millisecond-level) real-world dataset from the wireless networking domain. Our network data offers more drift scenarios, noise, and short-term changes, as compared to existing pre-trained datasets, thereby adding significant value to TSFMs. We have carefully addressed all identified weaknesses and questions raised during the review process. The reviewer's suggestions were invaluable and have helped us substantially improve the quality of the paper. We summarize our key clarifications and revisions as follows:

>R1:Q2, R3:W2:-We conducted an ablation study using two different fine-tuning strategies: (i) Head-only fine-tuning, (ii) Adapter-based fine-tuning. These strategies did not improve performance compared to the default TTM fine-tuning. The results are included in the Ablation Study section (Section 5, Page 9 and 10) of the revised draft.

>R1:Q3, R3:Q1:- We analyzed the performance of ARF and TSFMs under different sampling frequencies in a multivariate setting. Our results show that higher temporal resolution does not improve the performance of TSFMs, while ARF consistently outperforms them. The results are included in the Ablation Study section (Section 5, Page 9 and 10) of the revised draft.

>R2:W1:-To extend Chronos to the multivariate setting, we conducted both the zero-shot and fine-tuning experiments. ARF continues to outperform Chronos, reinforcing our earlier analysis. We have updated Table 4 (Section 4.4, Page 7) in the revised draft to reflect these new results.

>R2:W2:-We fixed the issue in the fine-tuning code, and reran the experiments with three seeds, and updated Table 4 (Section 4.4, Page 7) accordingly in the revised draft. We benchmarked both transformer-based models (Chronos and Lag-Llama) and a non-transformer model (TTM). While Chronos performs well in terms of MAE in the univariate setting, ARF consistently outperforms all these models in the multivariate setting.

>R2:W3,W4:-We conducted hyper-parameter tuning for the TSFM models, specifically for TTM and Lag-Llama. Despite this, ARF still outperforms both. We have added these results in Appendix (A.5.3, Page 21) in the revised version. We have also added the suggested online baseline, a simple incremental linear regression model (Section 4.1, Page 6), and validated that ARF still outperforms it. We have updated Table 4 and Fig.5. (Section 4.4, Page 7, 8).

>R2:W5:-We have clarified the data splitting procedure and seed handling (Section 4.3, Page 7:Model Parameters). Regarding seed control and repetition, we have identified and corrected issues in the original code and updated Table 4 (Section 4.4, Page 7) in the revised draft.

>R3:Q2:- We conducted an additional experiment using the zero-shot Chronos-base model. Although the improvement is modest over Chronos-small, ARF continues to outperform Chronos, indicating that the performance gap is not solely due to model scale. (Section A.5.3, Page 21 of revised draft.)

>R4:W1:-We have updated the title which more accurately reflects the focus on high-frequency network data. We also revised the Introduction (Section 1, Page 1) to clearly position our dataset as a valuable benchmark within the 5G wireless network domain, without generalizing to all types of high-frequency data.

>R4:W2:-We confirm using four most important features for main experiments, updated in Table 2 (Page 6). However, our dataset contains 47 features in totals. An additional experiments with the top 10 features are added in Appendix (Section A.5.1, Page 19). The same section also compares our feature count with existing pre-trained datasets, showing that our 47-feature dataset is well-suited for multivariate settings and training TSFMs (Page 18).

>R4:W3:-We have improved the resolution and clarity of all figures to ensure they are easily readable and visually clear in the revised draft.

>R1:Q1:-We clarified in the comments that this is an important direction but outside the scope of this paper, which focuses only on evaluating and fine-tuning existing TSFMs on a new domain (wireless networks) and new frequency (100 ms) as it has already been observed in TTM paper that resolution diversity and coverage is important. Addressed in detail in the R1:Q1 comment.

>We have also added potential use cases of this dataset to the Appendix (Section A.1, Page 12, 13) of the revised draft. These use cases demonstrate that the dataset is not limited to regression problems but also supports classification and anomaly detection tasks.

>We also include experiments on filtered subsets representing different mobility and traffic patterns to illustrate transfer learning potential in the Appendix (Section A.5.2, Page 18, 19, 20).

---

### Meta-Review · Area_Chair_CaLS · 2026-01-07

**Summary:**

The reviewers raised several concerns, the main ones are the following:

1. The analysis is largely focused on a filtered subset (static mobility + video streaming) of the wireless dataset (by reviewer qbv6 and NhZy). Reviewer 4iPt further questions the overstated scope of the paper.

2. Incomplete evaluation of Time Series Foundation Models (TSFMs) in the multivariate forecasting scenario (by reviewer qbv6). Reviewer uVpw and NhZy have concerns about TSFMs, where a modest, targeted sweep (e.g., context length, learning rate, adapter size) and a couple of online baselines (e.g., simple online linear/Kalman filters) would increase credibility.

3. Concerns about Chronos evaluation (reviewer uVpw).

4. Suspicious experimental results (by reviewer  uVpw), including many ± values in Table 4 are 0.0000, suggesting a single deterministic run or missing variance estimation; For Chronos, zero‑shot and fine‑tuned metrics are identical to four decimals (0.0313 MAE 0.0185). This is suspicious and suggests fine‑tuning may not have actually changed the model or was evaluated incorrectly.

**Reviewer Concerns:**

After the rebuttal, the authors did not address the concern 1 and concerns 2 above. For concern 3 and 4, the authors found some problems in the initial experiment and have corrected them in the revision.

Given the limited scope and immature experiment of the paper, at the current stage I cannot recommend acceptance. The authors are encouraged to take the suggestions of the reviewers for a future venue.

**Reviewer Scores:**

After the rebuttal, the reviewers will likely remain their scores of 4,4,4,2.

---

### Decision · Program_Chairs · 2026-01-26

Reject